# A novel RyR1-selective inhibitor prevents and rescues sudden death in mouse models of malignant hyperthermia and heat stroke

Toshiko Yamazawa [1,11 ✉], Takuya Kobayashi [2,11], Nagomi Kurebayashi [2,11], Masato Konishi[2], Satoru Noguchi [3], Takayoshi Inoue [4], Yukiko U. Inoue [4], Ichizo Nishino [3], Shuichi Mori[5], Hiroto Iinuma[5], Noriaki Manaka[5], Hiroyuki Kagechika[5], Arkady Uryash [6], Jose Adams[6], Jose R. Lopez[7], Xiaochen Liu[8], Christine Diggle[8], Paul D. Allen[8], Sho Kakizawa[9], Keigo Ikeda[10], Bangzhong Lin [10], Yui Ikemi [10], Kazuto Nunomura [10], Shinsaku Nakagawa[10], Takashi Sakurai [2] & Takashi Murayama [2 ✉]

Mutations in the type 1 ryanodine receptor (RyR1), a $Ca^{2+}$ release channel in skeletal muscle, hyperactivate the channel to cause malignant hyperthermia (MH) and are implicated in severe heat stroke. Dantrolene, the only approved drug for MH, has the disadvantages of having very poor water solubility and long plasma half-life. We show here that an oxolinic acid-derivative RyR1-selective inhibitor, 6,7-(methylenedioxy)-1-octyl-4-quinolone-3-car-boxylic acid (Compound 1, Cpd1), effectively prevents and treats MH and heat stroke in several mouse models relevant to MH. Cpd1 reduces resting intracellular $Ca^{2+}$, inhibits halothane- and isoflurane-induced $Ca^{2+}$ release, suppresses caffeine-induced contracture in skeletal muscle, reduces sarcolemmal cation influx, and prevents or reverses the fulminant MH crisis induced by isoflurane anesthesia and rescues animals from heat stroke caused by environmental heat stress. Notably, Cpd1 has great advantages of better water solubility and rapid clearance in vivo over dantrolene. Cpd1 has the potential to be a promising candidate for effective treatment of patients carrying RyR1 mutations.

[1] Department of Molecular Physiology, The Jikei University School of Medicine, Tokyo, Japan. [2] Department of Pharmacology, Juntendo University School of Medicine, Tokyo, Japan. [3] Department of Neuromuscular Research, National Institute of Neuroscience, National Center of Neurology and Psychiatry, Tokyo, Japan. [4] Department of Biochemistry and Cellular Biology, National Institute of Neuroscience, National Center of Neurology and Psychiatry, Tokyo, Japan. [5] Institute of Biomaterials and Bioengineering, Tokyo Medical and Dental University, Tokyo, Japan. [6] Department of Neonatology, Mount Sinai Medical Center, Miami, FL, USA. [7] Department of Research, Mount Sinai Medical Center, Miami, FL, USA. [8] Leeds Institute of Biomedical & Clinical Sciences, School of Medicine, University of Leeds, St James's University Hospital, Leeds, UK. [9] Department of Biological Chemistry, Graduate School of Pharmaceutical Sciences, Kyoto University, Kyoto, Japan. [10] Center for Supporting Drug Discovery and Life Science Research, Graduate School of Pharmaceutical Science, Osaka University, Suita, Japan. [11] These authors contributed equally: Toshiko Yamazawa, Takuya Kobayashi, Nagomi Kurebayashi. ✉email: toshiko1998@jikei.ac.jp; takashim@juntendo.ac.jp

The type 1 ryanodine receptor (RYR1; MIM# 180901) is a $Ca^{2+}$-release channel in the sarcoplasmic reticulum (SR) of skeletal muscle that plays a central role in muscle contraction[1,2]. During excitation–contraction (E–C) coupling, RyR1 releases $Ca^{2+}$ through physical association with the sarcolemmal slow voltage gated $Ca^{2+}$ channel (dihydropyridine receptor, DHPR) during depolarization of T-tubule membrane. This mechanism is referred to as depolarization-induced $Ca^{2+}$ release (DICR)[3,4]. The RyR1 channel can also be directly activated by $Ca^{2+}$, which is referred to as $Ca^{2+}$-induced $Ca^{2+}$ release (CICR)[5,6], although its importance in the function is much lower than RyR2 and RyR3[7].

Genetic mutations in the RYR1 gene are associated with malignant hyperthermia (MH; MIM# 145600)[8]. MH is a life-threatening disorder characterized by skeletal muscle rigidity and elevated body temperature in response to all volatile anesthetics such as halothane or isoflurane[9]. MH mutations are thought to hyperactivate CICR, which causes massive $Ca^{2+}$ release by halogenated anesthetics under resting conditions[6,10,11]. Heat stroke is a medical emergency with a high body temperature and altered mental status, which is triggered by exercise or environmental heat stress[12,13]. It has been reported that MH mutations in the RYR1 gene are also implicated in some heat stroke[14–16].

The only drug approved for ameliorating the symptoms of MH is dantrolene[17,18]. Dantrolene was first developed in 1960s as a muscle relaxant[19] and was later shown to prevent $Ca^{2+}$ release by direct interaction with RyR1[20,21]. However, dantrolene has several disadvantages for clinical use: the main disadvantage is its poor water solubility which makes rapid preparation difficult in emergency situations[22]. In addition, dantrolene has long plasma half-life ($t_{1/2} = 12$ h), which causes long-lasting side effects such as muscle weakness[23]. To date, no alternative drugs improving these disadvantages have been developed over 40 years since the first discovery.

We have recently identified oxolinic acid as a RyR1-selective inhibitor with better water solubility using efficient high-throughput screening platform for RyR1 inhibitors[24,25]. We synthesized a series of oxolinic acid derivatives by structural development, and successfully identified 6,7-(methylenedioxy)-1-octyl-4-quinolone-3-carboxylic acid (Compound 1, Cpd1) with extremely high potency which is comparable to dantrolene in in vitro study[26]. In this study, we tested whether Cpd1 has therapeutic effects in treating the MH crisis in a MH mouse model carrying RYR1-p.R2509C, a corresponding mutation (p.R2508C) in humans[27,28] and two other MH mouse models, RYR1-p.R163C[29] and RYR1-p.G2435R[30]. We found that Cpd1 effectively prevents and reverses a fulminant MH crisis triggered by isoflurane anesthesia. The drug also treats mice with heat stroke caused by environmental heat stress. A notable finding is that Cpd1 is rapidly metabolized in mice. Our results provide crucial evidence for Cpd1 as a RyR1 inhibitor that may prove to be clinically useful.

## Results
**Generation of MH model mice carrying RYR1-p.R2509C mutation.** A MH mouse model carrying p.R2509C mutation in the RYR1 gene (RYR1-p.R2509C) was created using CRISPR/Cas9 system (see Methods, Fig. 1a). An additional Xba I site was introduced to screen the genotype by PCR-RFLP. The homozygous RYR1-p.R2509C mice died in late embryonic stage, but heterozygous RYR1-p.R2509C (R2509C) mice grew normally and were as fertile as wild type (WT) (Fig. 1b).

It has previously been shown when MH susceptible mice, swine or humans with mutations in the RYR1 gene are exposed to volatile anesthetics, it causes a MH crisis and if untreated can lead

to death[29–34]. When anesthetized by isoflurane, nearly 80% (68 out of 83) heterozygous R2509C mice rapidly increased their rectal temperature to over 40 °C ($42.3 \pm 1.3$ °C, $n = 16$) (Fig. 1c, d), exhibited muscle rigidity (Fig. 1c, inset) and died within 90 min ($64 \pm 30$ min, $n = 16$) from start of exposure (Fig. 1e). In the remaining R2509C mice, exposure to isoflurane caused an increase in rectal temperature but it did not exceed 39 °C. In contrast, WT mice showed no elevation in rectal temperature in response to isoflurane (Fig. 1c, d). Elevation of rectal temperature looks two-phase reaction; it rose slowly at first and then surged around 39 °C (Fig. 1c). Interestingly, time from rectal temperature of 39 °C to death was quite similar among all individual mice ($16 \pm 4$ min, $n = 16$) (Fig. 1e). We further investigated sex difference in response to isoflurane. More male mice (90%) succumbed to isoflurane than females (65%) (Fig. 1f). Among the mice that succumbed, there was no significant sex difference in the rectal temperature (Fig. 1g) and time to death (Fig. 1h, i). These findings suggest that R2509C mice can reproduce human MH symptoms in response to exposure to volatile anesthetics and thus are a useful model for test of Cpd1.

**Effect of Cpd1 on intracellular $Ca^{2+}$ homeostasis in the isolated single muscle cells.** Cpd1 is a derivative of oxolinic acid with an octyl group at nitrogen atom (oxolinic acid has an ethyl group)[26]. This modification increased the potency for inhibiting the RyR1 channel nearly 70-fold in a HEK293 cell-based assay ($IC_{50}$ values from 810 nM for oxolinic acid to 12 nM for Cpd1). Since Cpd1 is a free acid, it is virtually water insoluble. We therefore prepared its sodium salt, which neutralizes the carboxyl group and significantly increases water solubility (Fig. 2a). The Cpd1 sodium salt exhibited good solubility in normal saline (0.9% NaCl) as well as pure water and glucose solution (Table 1). In contrast, dantrolene sodium salt was soluble in pure water and glucose solution but hardly soluble in normal saline, which is in accord with previous studies[35].

We initially tested the effect of Cpd1 in in vitro experiments using enzymatically isolated single flexor digitorum brevis (FDB) muscle cells from the R2509C mice. Abnormal resting intracellular $Ca^{2+}$ ($[Ca^{2+}]_i$) homeostasis in skeletal muscle cells is one of the most common features associated with MH in susceptible humans[36], swine[31,32], and MH model mice[29,30,33,34]. Muscle cells were loaded with a fluorescent $Ca^{2+}$ indicator, Cal520, and $[Ca^{2+}]_i$ was fluorometrically monitored during exposure of the cells to halothane, which is used for clinical MH testing[37]. Whereas halothane up to 0.1% (v/v) did not change $[Ca^{2+}]_i$ in WT cells, it caused increase in $[Ca^{2+}]_i$ in R2509C cells in a dose-dependent manner (Fig. 2b, c). Pretreatment of R2509C cells with Cpd1 (0.1 μM) reduced resting $[Ca^{2+}]_i$ (Supplementary Fig. 1a, b) and completely abolished halothane-induced increases in $[Ca^{2+}]_i$ (Fig. 2b, c). We then used the ratiometric $Ca^{2+}$ indicator fura-2 to estimate the unstimulated resting $[Ca^{2+}]_i$ and whether or not it could be modified by Cpd1. At rest fura-2 $F_{340}/F_{380}$ ratios were significantly higher in R2509C cells than in WT and Cpd1 reduced $[Ca^{2+}]_i$ in R2509C cells in a dose-dependent manner but had no significant effect on WT (Fig. 2d). Similar to the effects of halothane, exposure to isoflurane increased $[Ca^{2+}]_i$ in R2509C but not in WT cells and Cpd1 completely prevented this effect (Fig. 2e, f).

It has been shown that sarcolemmal cation influx is accelerated in skeletal muscle of several MH model mice compared with WT mice[38,39]. This is due to enhanced sarcolemmal $Ca^{2+}$ entry in response to chronic reduced SR $Ca^{2+}$ load. We therefore measured resting $Mn^{2+}$ quench of fura-2 fluorescence. Unexpectedly, rate of $Mn^{2+}$ quench in R2509C muscles was not different from WT and in this model Cpd1 had no effect (Supplementary Fig. 2a, b).

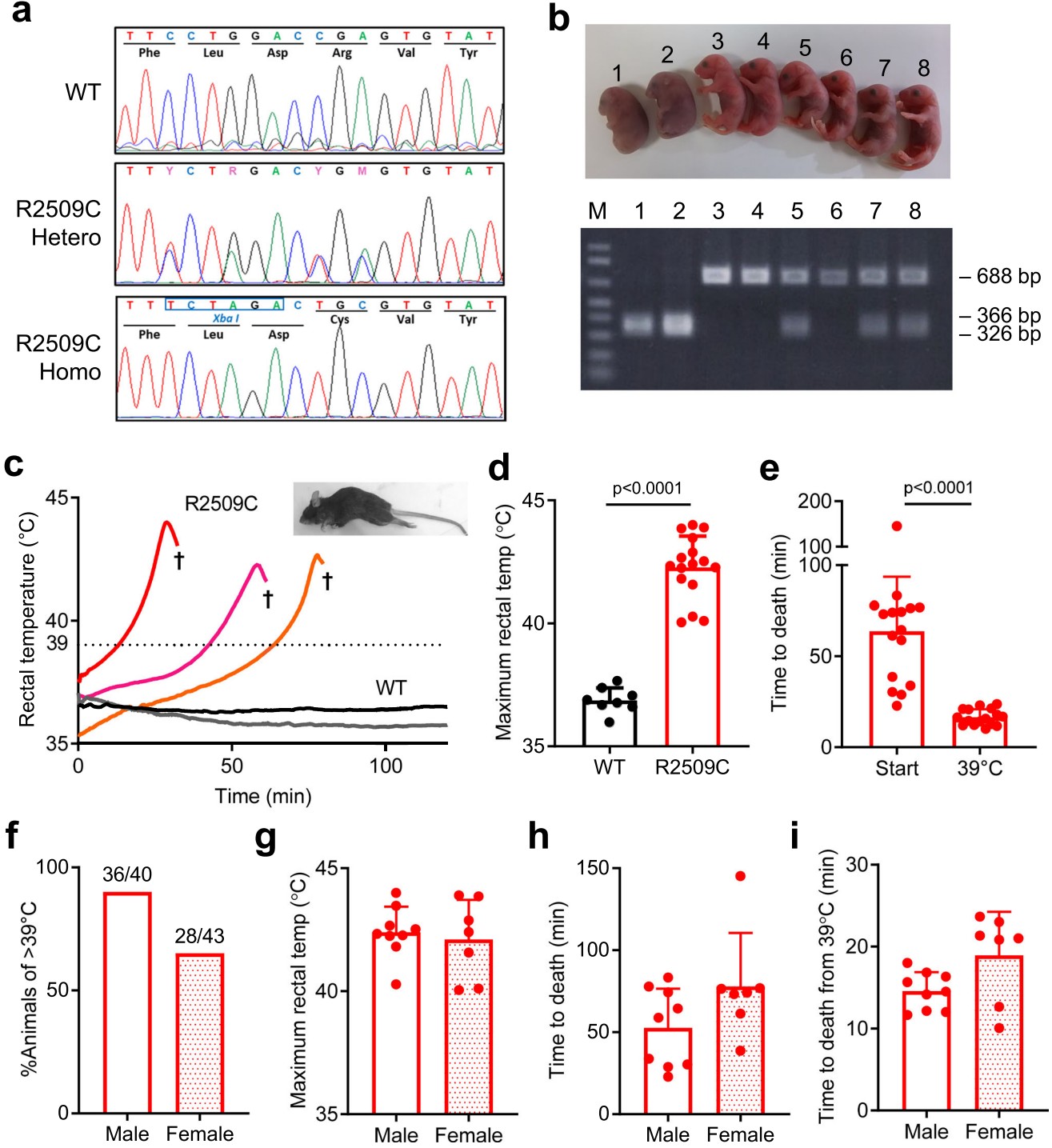

**Effect of Cpd1 on contraction force in isolated muscles**. The In Vitro Contracture Test (IVCT) using muscle biopsies is widely used for diagnosis of MH susceptibility in humans[40]. A subject is considered MH positive if the muscle exhibited a greater contracture at a lower dose of caffeine or halothane. We therefore tested the effect of Cpd1 on caffeine-induced contracture in soleus muscles isolated from R2509C mice. WT muscles showed only minimal contracture in responses to caffeine up to 20 mM (Fig. 3a). In contrast, R2509C muscles exhibited dose-dependent contracture during exposure to caffeine and pretreatment with 3 µM Cpd1 decreased caffeine-induced contracture tension of R2509C muscles at 20 mM caffeine (Fig. 3b).

It has been reported that soleus muscles from MH model mice exhibit heat-induced increase in the basal force at temperatures of 35 °C or higher[33]. Basal tension at 42 °C was significantly increased in R2509C solei compared to WT muscles (Fig. 3c, d) and pretreatment with 3 µM Cpd1 significantly reduced the heat-induced contracture in R2509C muscles.

Dantrolene is known to suppress twitch and tetanic contractions in both normal and MH muscles[41,42]. We tested whether Cpd1 affects twitch and tetanic contractions and whether or not there was a difference in the Cpd1 effect between MH and WT. Untreated soleus muscles from both WT and R2509C mice showed similar twitch and tetanic contraction force and

**Fig. 1 Generation and characterization of RYR1-p.R2509C (R2509C) mice.** R2509C mice were generated by CRISPR/Cas9 system using single-stranded donor oligonucleotide carrying Xba I restriction site (TCTAGA). **a** Sequences of an amplified DNA fragment from a tail of WT, heterozygous (Hetero), and homozygous (Homo) offspring. Sequencing confirms the presence of both the R2509C mutation and a Xba I site on both alleles in homozygous and on one allele in heterozygous offspring. **b** Gross morphology and allele-specific PCR-RFLP analysis of WT, heterozygous, and homozygous embryos at E19 day. Top, Heterozygous embryos (No. 5, 7, and 8) showed no apparent differences from WT embryos (No. 3, 4, and 6), but homozygous embryos (No. 1 and 2) were dead. Bottom, presence of a single 688 bp band denotes WT mice, 366 bp and 326 bp bands (not well separated in this gel) denote homozygous mice, and all three bands denote heterozygous mice. Typical results of three independent experiments are shown. **c** Change in rectal temperature of mice after isoflurane challenge. Heterozygous R2509C mice but not WT mice exhibited elevation in rectal temperature and died by fulminant MH crisis (†). Inset, R2509C mice responded with full body contractions as reflected in arching of their backs and extension of their legs. **d** Maximum rectal temperature of WT ($n = 8$) and R2509C ($n = 16$) mice during isoflurane challenge. Data are shown as means ± SD and were analyzed by two-tailed unpaired $t$-test. **e** Time to death from the start of isoflurane challenge or from rectal temperature of 39 °C of R2509C mice. Data are expressed as means ± SD ($n = 16$) and were analyzed by two-tailed unpaired $t$-test. **f** Responsiveness to isoflurane between male and female mice. Whereas 90% (36 out of 40) of male mice increased the body temperature to >39 °C, only 65% (28 out of 43) of female mice exceeded 39 °C. **g-i** Comparison between male ($n = 9$) and female ($n = 7$) R2509C mice during isoflurane challenge; maximum rectal temperature (**g**), time to death from isoflurane challenge (**h**), and time to death from the time rectal temperature reached 39 °C during isoflurane challenge (**i**). Data are shown as means ± SD ($n = 16$) and were analyzed by two-tailed unpaired $t$-test. There was no statistically significant difference between male and female mice. Source data are provided as a Source Data file.

treatment with 3 μM Cpd1 significantly reduced both twitch and tetanic contraction force in both groups (Fig. 3e–g). The effect was stronger on twitch (72% inhibition) (Fig. 3f) than on tetanic (35% inhibition at 100 Hz) (Fig. 3g) contractions, which is similar to dantrolene[41].

**Effect of Cpd1 on in vivo isoflurane challenge.** We next tested whether Cpd1 can prevent or treat MH episodes in R2509C mice in vivo. We first tested the preventive effect of Cpd1 on MH episodes in response to isoflurane exposure. We used male mice for the experiments, since male mice were more sensitive to isoflurane than female mice (see Fig. 1f). Doses of either 3 mg/kg or 10 mg/kg of Cpd1 solubilized in normal saline were administered i.p. 10 min before isoflurane challenge. The 3 mg/kg dose did not prevent rise in rectal temperature and only 1 out of 6 mice survived (Fig. 4a–c). Time to death for mice treated at 3 mg/kg (47 ± 15 min, n = 5) was not significantly different than that for control mice (53 ± 24 min, n = 9). However, with the 10 mg/kg dose, Cpd1 successfully prevented any rise in rectal temperature (Fig. 4a, b). All the treated mice survived 90 min after induction of anesthesia (Fig. 4c) and they behaved normally for at least 24 h after recovery from anesthesia.

We next tested whether Cpd1 could rescue mice from MH episodes. R2509C mice (both males and females) were anesthetized with isoflurane and the drug was administered i.p. when rectal temperature reached 39 °C. The 3 mg/kg dose was able to reduce rectal temperature by −0.03 ± 0.81 °C at 10 min after administration, compared to an increase of 1.79 ± 0.82 °C in controls, but rectal temperature started to rise again ~30 min after administration of the drug in some mice (Fig. 4d, e). The 10 mg/kg dose decreased body temperature by −0.64 ± 0.29 °C at 10 min after administration and maintained it at that level until the experiment was terminated 60 min after administration of the drug. At that time 60% of the mice in the 3 mg/kg group and 100% of the mice in the 10 mg/kg group survived (Fig. 4f) and all the survivors behaved normally for at least 24 h after experiments.

**Effect of Cpd1 on in vivo environmental heat stress challenge.** It has been shown that environmental heat stress causes an increase in body temperature and death in other MH model mice[29,30,33,34]. We therefore tested whether R2509C mice exhibit heat stroke triggered by environmental heat stress. Mice were anesthetized using intravenous anesthetics and transferred to a 35 °C test chamber. Whereas the rectal temperature of WT mice was maintained at a relatively constant level around 37 °C (37.4 ±

0.8 °C, $n = 7$), the rectal temperature of R2509C mice steadily increased to 41.4 ± 1.4 °C ($n = 13$) (Supplementary Fig. 3a). Almost all the mice responded to heat and died (Supplementary Fig. 3b). Although the time from start of heat stress to death largely varied among individual mice, the variation in time to death after reaching body temperatures of 38 °C (36 ± 13 min, $n = 13$) or 39 °C (22 ± 8 min, $n = 12$) became smaller (Supplementary Fig. 3c).

Using this model, we initially tested the preventive effect of Cpd1 on the heat stroke. Pretreatment of mice with Cpd1 (10 mg/kg) slowed the rate of temperature rise (Fig. 5a), but maximum rectal temperature did not change (Fig. 5b) and none of the treated mice (0 out of 4) survived 120 min of heat stress (Fig. 5c). However, time to death for the treated mice (98 ± 17 min, $n = 4$) was significantly prolonged compared to the control mice (74 ± 21 min, $n = 13$) (Fig. 5d).

We next tested whether Cpd1 could rescue mice from environmental heat stroke. The R2509C mice were incubated in the test chamber and the drug was administered i.p. when their rectal temperature reached 39 °C. All the untreated mice died within 30 min of the time of injection (time to death, 22 ± 8 min, $n = 13$). The 3 mg/kg Cpd1 dose slowed rise in the body temperature post injection (Fig. 5e, f) and 3 out of 5 mice survived 60 min after administration (Fig. 5g). The 10 mg/kg Cpd1 dose showed more pronounced effect: it suppressed rise in the body temperature and 10 out of 11 mice survived for 60 min after administration. After removing from the heat stress environment, the survivors behaved normally for at least 24 h after being and regaining consciousness. When the drug was administered at body temperature of 38 °C, after which most untreated mice died within 60 min (time to death, 36 ± 13 min, $n = 13$), both the 3 and 10 mg/kg Cpd1 doses strongly suppressed the continued rise in the body temperature (Supplementary Fig. 4a, 4b) and almost all mice survived 60 min after its administration (Supplementary Fig. 4c).

**Pharmacokinetics of Cpd1.** We observed transient effects with the 3 mg/kg dose of Cpd1 in rescue experiments in both in vivo isoflurane and heat stress challenges with R2509C mice (see Figs. 4d and 5e). This implies fast metabolism or excretion of the drug in mice. We therefore measured plasma concentration of Cpd1 after i.p. administration. Cpd1 rapidly declined in mouse blood following first order kinetics with $t_{1/2}$ of ~8 min for both the 3 mg/kg and 10 mg/kg dose (Fig. 6a). There was no sex difference in the kinetics of Cpd1 (Supplementary Fig. 5a, b). Since Cpd1 has hydrophobic moiety of octyl group (see Fig. 2), it can be metabolized in the liver. In vitro drug metabolism assay using

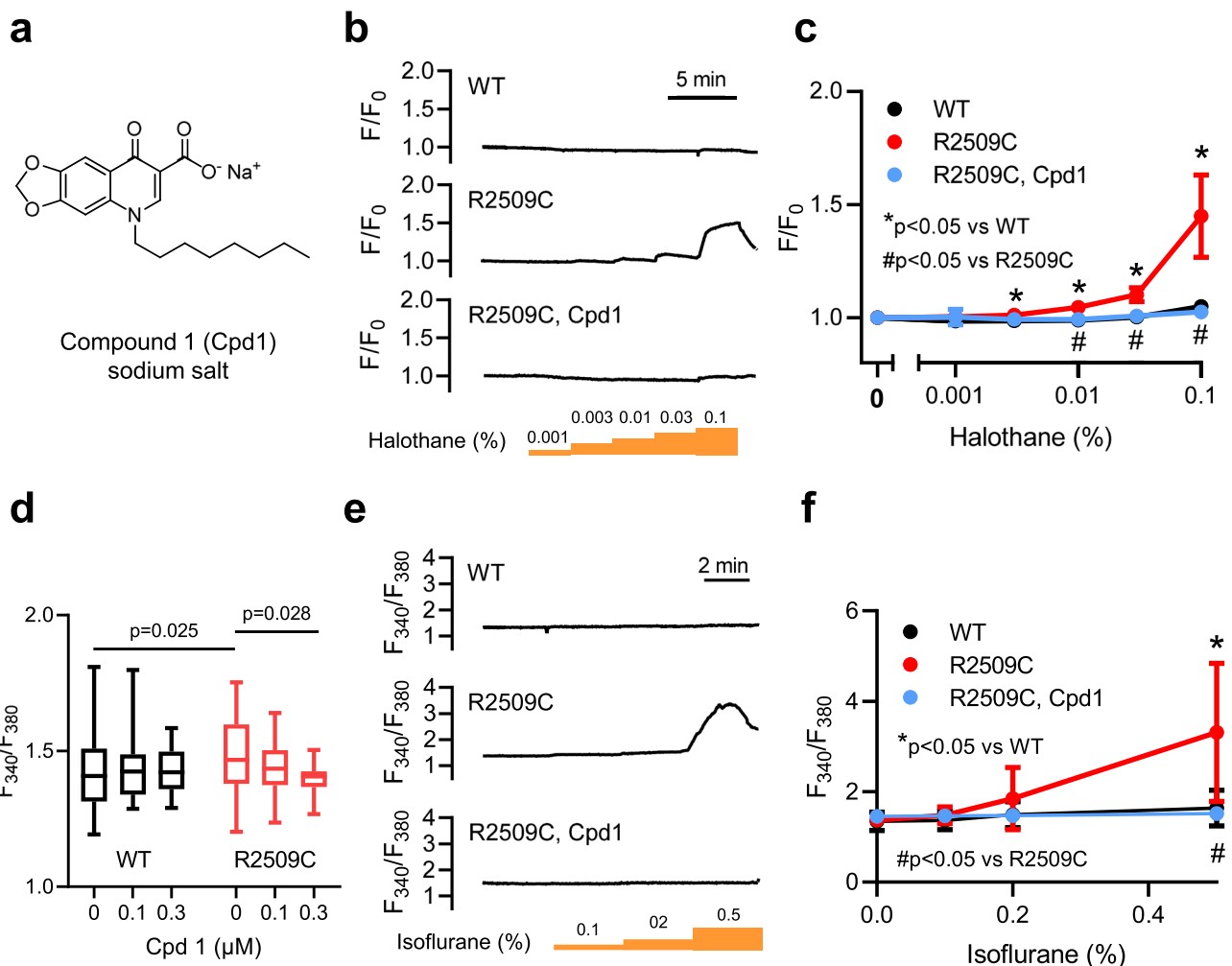

**Fig. 2 Intracellular Ca²⁺ signals in single FDB muscle cells isolated from WT and R2509C mice. a** Structure of Cpd1 sodium salt. **b** Representative Ca²⁺ signals in the isolated FDB muscle cells from WT and R2509C mice at various concentrations of halothane in the absence (WT and R2509C) and presence of 0.1 μM Cpd1 (R2509C, Cpd1). Fluorescence Ca²⁺ signals (F) were detected with Cal520 and normalized to the initial value ($F_0$). **c** Average resting Ca²⁺ signals at various concentrations of halothane and effects of 0.1 μM Cpd1. Data are shown as means ± SD (WT: $n = 4$, R2509C: $n = 7$, R2509C, Cpd1: $n = 5$) and were analyzed by one-way ANOVA with Tukey's test. **d** Average Ca²⁺ signals obtained with fura-2 at rest and in the presence of Cpd1. Data are shown as box-whisker plots, with the median for all subjects shown as the center line, the box representing the 25–75 percentile, and the lines showing the range of the data (WT, 0 μM: $n = 34$; WT, 0.1 μM: $n = 21$; WT, 0.3 μM: $n = 16$; R2509C, 0 μM: $n = 45$; R2509C, 0.1 μM: $n = 28$; R2509C, 0.3 μM: $n = 17$) and were analyzed by two-way ANOVA with Tukey's test. **e** Representative fura-2 ratio signals in isolated FDB muscle cells from WT and R2509C mice at various concentrations of isoflurane in the absence (WT and R2509C) and presence of 0.1 μM Cpd1 (R2509C, Cpd1). **f** Average resting Ca²⁺ signals at various concentrations of isoflurane and effects of 0.1 μM Cpd1. Data are shown as means ± SD (WT: $n = 11$; R2509C: $n = 8$ or 10, 2 cells died after exposure to 0.5% isoflurane; R2509C, Cpd1: $n = 8$) and were analyzed by one-way ANOVA with Tukey's test. Source data are provided as a Source Data file.

**Table 1 Solubilities of Cpd1 sodium salt and dantrolene sodium salt in solutions for injection.**

|  | Cpd1 sodium salt | Dantrolene sodium salt |
|---|---|---|
| Normal saline (0.9% NaCl) | 849 ± 58 (4) | 25.7 ± 2.1 (3) |
| Pure water | 1360 ± 220 (4) | 762 ± 67 (3) |
| Glucose solution (5%) | 351 ± 14 (4) | 744 ± 22 (3) |

*Values are represented in μg/mL.
**Number of tests (n) are indicated in parenthesis.

mouse liver microsomes demonstrated that Cpd1 was reduced to 20% and 5% of the original amount at 10 min and 60 min, respectively (Fig. 6b). Cpd1 was metabolized more slowly by human liver microsomes; it was reduced to 20% at 60 min.

The duration of the effect of Cpd1 in mice was also evaluated using measurements of in vivo muscle strength, since we showed that the drug inhibited twitch and tetanic contractions in isolated skeletal muscle (see Fig. 2). Grip strength tests demonstrated that muscle strength of WT mice was reduced by 15% 10 min after administration of 10 mg/kg Cpd1, and that this deficit was almost completely recovered at 60 min, suggesting short duration of the medicinal effect in vivo (Fig. 6c). No sex nor genotype differences were observed in the effect of Cpd1 on in vivo muscle contraction strength (Supplementary Fig. 6a–c).

The above results indicate that transient effects of Cpd1 in the isoflurane challenge and heat challenge experiments are due to rapid metabolism of the drug in mice. In fact, second and third administrations of the 3 mg/kg dose when rectal temperature reached 39 °C successfully decreased the rectal temperature (Supplementary Fig. 7). We therefore tested whether repeated administrations of Cpd1 could prevent mice from the crisis. Eight

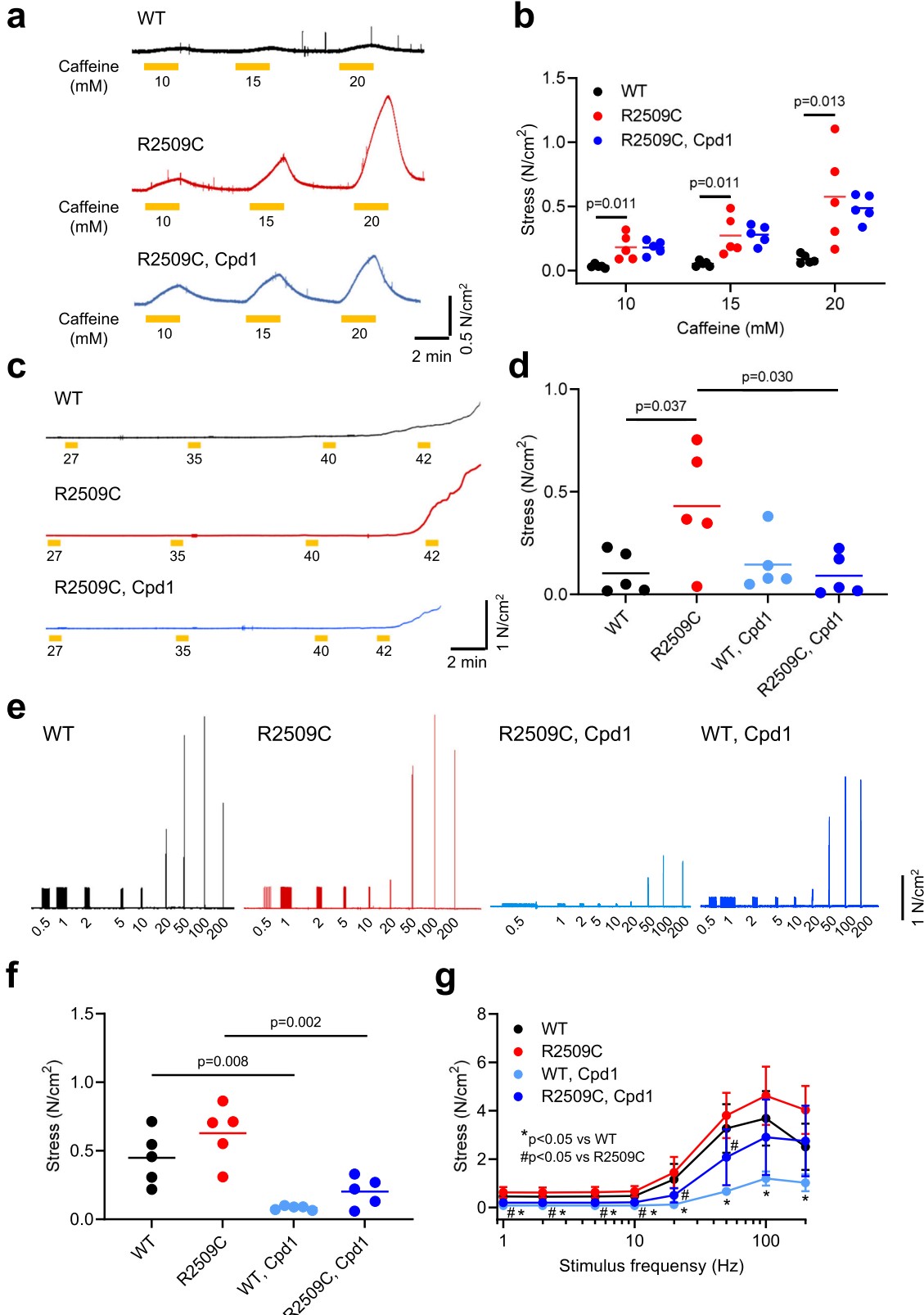

administrations of 1 mg/kg Cpd1 at 10 min intervals successfully prevented rise in the rectal temperature by exposure either to isoflurane (Fig. 6d) or to heat stress (Fig. 6e) and all the mice survived the full 90 min after initiating either challenge (Fig. 6f). Pharmacokinetic analysis of Cpd1 after repeated administrations

revealed that Cpd1 rapidly declined from mouse blood as was the case of single administration (Fig. 6g). Interestingly, $t_{1/2}$ was slightly longer in female ($14.8 \pm 2.1$ min, $n = 3$) than in male ($9.6 \pm 2.7$ min, $n = 3$), whereas $C_{max}$ was significantly higher in males ($0.67 \pm 0.11$ μg/mL, $n = 3$) than in females ($0.15 \pm 0.11$ μg/mL,

**Fig. 3 Contractile function in soleus muscles from WT and R2509C mice. a** Typical traces of basal stress after exposure to caffeine. Caffeine at indicated concentrations was applied for 2 min. Soleus muscles from R2509C mice exhibit much greater increase in basal stress after exposure to caffeine than those from WT mice. Pretreatment of muscles with Cpd1 (3 μM) effectively reduces the increase in basal stress caused by caffeine in R2509C muscles. **b** Dose dependence of the increase in basal stress in response to caffeine exposure. Data are shown as means with individual data points ($n = 5$ for each group) and were analyzed by one-way ANOVA with Tukey's test. **c** Typical traces of basal stress with increased temperature. **d** The average values of basal stress for 30 s after exposure to 42 °C. Data are shown as means with individual data points ($n = 5$ for each group) and were analyzed by one-way ANOVA with Tukey's test. **e** Typical traces of tension development by electrical stimuli. Muscles were stimulated with 0.5 Hz pulses to evoke twitch and at 1 min intervals with subsequent 20 trains of 1, 2, 5, 10, 20, 50, 100, and 200 Hz pulses. **f** Summary of twitch tension. Data are shown as means with individual data points ($n = 5$ for each group) and were analyzed by one-way ANOVA with Tukey's test. No significant difference was observed between WT and R2509C muscles. Cpd1 (3 μM) significantly reduces twitch tension in both WT and R2509C muscles. **g** Stress–frequency relationship. Data are shown as means ± SD ($n = 5$ for each group) and were analyzed by one-way ANOVA with Tukey's test. No significant difference was observed between WT and R2509C muscles. Cpd1 (3 μM) significantly reduces tetanic stress in both groups. Source data are provided as a Source Data file.

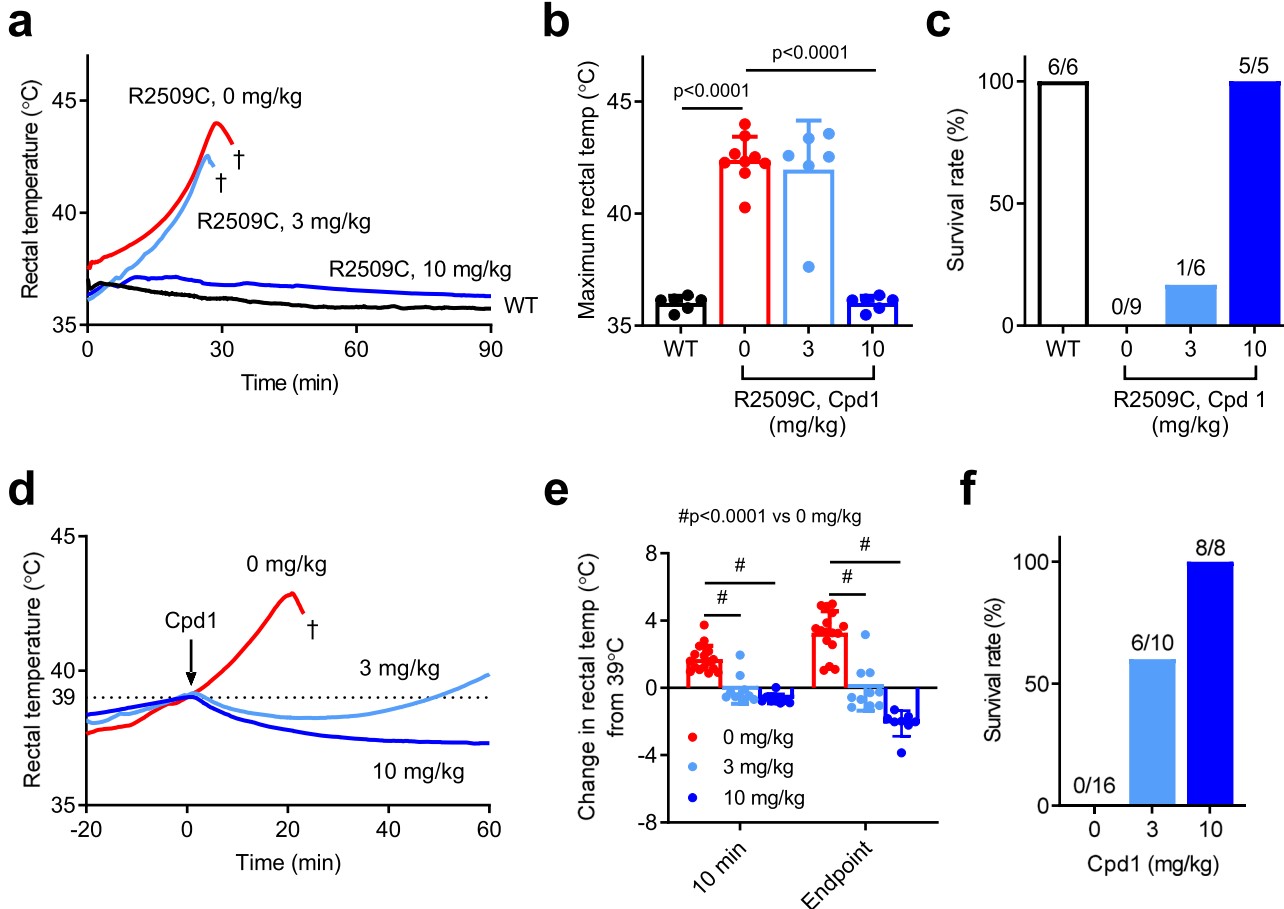

**Fig. 4 In vivo isoflurane challenge in WT and R2509C mice. a–c** Preventive effects of Cpd1. **a** Time course of rectal temperature in mice after anesthesia by isoflurane. Cpd1 (0, 3, 10 mg/kg) was administered i.p. to R2509C mice 10 min before anesthesia. †, death by fulminant MH crisis. **b** Maximum rectal temperature. Data are shown as means ± SD (WT: $n = 6$; R2509C, 0 mg/kg: $n = 9$; R2509C, 3 mg/kg: $n = 6$; R2509C, 10 mg/kg: $n = 6$) and were analyzed by one-way ANOVA with Tukey's test. **c** Survival rate of mice after 90 min of anesthesia. Administration of Cpd1 at 10 mg/kg but not 3 mg/kg prevented R2509C mice from fulminant MH crisis. **d–f** Rescue effects of Cpd1 on fulminant MH crisis in R2509C mice. **d** Time course of rectal temperature in mice after administration of Cpd1 during anesthesia. Mice were anesthetized by isoflurane and Cpd1 (0, 3, 10 mg/kg) was administered i.p. when their body temperature reached 39 °C (arrow). †, death by fulminant MH crisis. **e** Change in rectal temperature of R2509C mice from 39 °C at 10 min after the administration of Cpd1 or saline control and the endpoint (60 min after administration or just before death). Data are shown as means ± SD (0 mg/kg: $n = 16$; 3 mg/kg: $n = 10$; 10 mg/kg: $n = 8$) and were analyzed by one-way ANOVA with Tukey's test. **f** Survival rate of R2509C mice 60 min after administration of Cpd1 or saline control. Source data are provided as a Source Data file.

$n = 3$). Muscle weakness was transiently observed after the repeated administrations in both male and female, but it returned to normal by 60 min (Fig. 6h).

**Effect of Cpd1 on *RYR1*-p.R163C MH model mice.** We demonstrated that Cpd1 prevented and reversed MH and heat stroke

in R2509C mice. To test whether this is also the case with other MH model mice carrying different mutations, we examined the effects of Cpd1 on heterozygous *RYR1*-p.R163 (R163C) mice[29] using in vivo measurements of $[Ca^{2+}]_i$ with $Ca^{2+}$-selective microelectrodes[38]. Resting $[Ca^{2+}]_i$ was threefold higher in R163C muscle (330 ± 43 nM, $n = 17$) compared to WT muscle (122 ± 3 nM, $n = 20$) (Fig. 7a). Administration of Cpd1 i.p. effectively reduced $[Ca^{2+}]_i$ in R163C

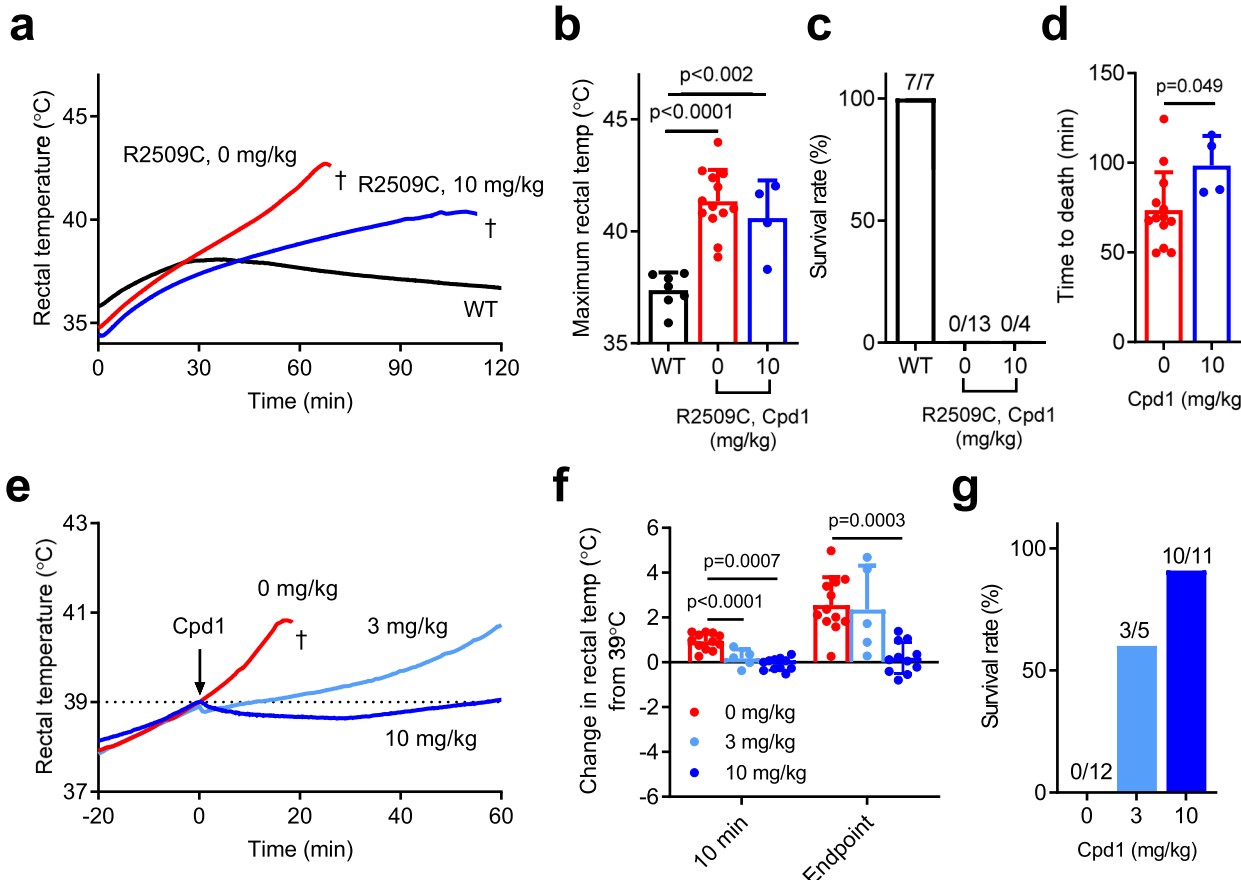

**Fig. 5 In vivo heat stress challenge in WT and R2509C mice.** Anesthetized mice were placed in a test chamber prewarmed at 35 °C. **a–c** Cpd1 (10 mg/kg) was administered 10 min before heat stress challenge. **a** Time course of the change of rectal temperature of mice. **b** Maximum rectal temperature. Data are shown as means ± SD (WT: $n = 7$; R2509C, 0 mg/kg: $n = 13$; R2509C, 10 mg/kg: $n = 4$) and were analyzed by one-way ANOVA with Tukey's test. **c** Survival rate of mice 120 min after heat stress challenge. **d** Time to death from start of heat stress challenge. Data are shown as means ± SD (0 mg/kg: $n = 13$; 10 mg/kg: $n = 4$) and were analyzed by two-tailed unpaired $t$-test. **e–g** Rescue effects of Cpd1 on heat stroke in R2509C mice. **e** Time course of rectal temperature of mice. Cpd1 (0, 3, 10 mg/kg) was administered i.p. when their body temperature reached 39 °C (arrow). †, death by heat stroke. **f** Change in the rectal temperature from 39 °C at 10 min after the administration of Cpd1 or saline control and the endpoint (60 min after administration or just before death). Data are shown as means ± SD (0 mg/kg: $n = 12$; 3 mg/kg: $n = 5$; 10 mg/kg: $n = 11$) and were analyzed by one-way ANOVA with Tukey's test. **g** Survival rate of R2509C mice 60 min after administration of Cpd1 or saline control. Source data are provided as a Source Data file.

muscle in a dose-dependent manner. Cpd1 also reduced $[Ca^{2+}]_i$ in WT muscles, but the effect was significantly smaller. When mice were exposed to 1.5% (v/v) isoflurane, $[Ca^{2+}]_i$ rapidly increased in untreated R163C muscles ($1226 \pm 154$ nM, $n = 19$), indicating a MH episode (Fig. 7b, c). Administration of isoflurane had no effect on $[Ca^{2+}]_i$ in WT muscles ($121 \pm 4$ nM, $n = 18$) (Fig. 7c). Pretreatment with Cpd1 suppressed increase in $[Ca^{2+}]_i$ in R163C muscle in a dose-dependent manner (Fig. 7b, c) and effectively prevented the MH episode and death at doses of 5 and 10 mg/kg (Fig. 7d). When Cpd1 at 10 mg/kg was administered to R163C mice 15 min after induction of isoflurane anesthesia, it was able to abort the MH crisis and rapidly decreased $[Ca^{2+}]_i$ to the levels measured in WT muscle (Fig. 7e, f).

To validate in vivo experiments, we tested the effect of Cpd1 in FDB muscle fibers isolated from the R163C mice. Similar to the in vivo findings, $[Ca^{2+}]_i$ was significantly higher in quiescent R163C fibers ($326 \pm 27$ nM, $n = 16$) compared to WT fibers ($121 \pm 3$ nM, $n = 20$) and Cpd1 reduced $[Ca^{2+}]_i$ in R163C fibers in a dose-dependent manner to the level of WT fibers. (Supplementary Fig. 8).

**Effect of Cpd1 on *RYR1*-p.G2435R MH model mice.** We also tested the effect of Cpd1 on a third MH model using homozygous *RYR1*-p.G2435R (G2435R) mice[30]. G2435R mice exhibit severe

heat stroke by environmental heat stress. When unanesthetized restrained G2435R mice were exposed to an ambient temperature of 38 °C, their rectal temperature rapidly increased and reached 40 °C within 20 min (the experiment was terminated at this point) (Fig. 8a). Pretreatment of G2435R mice with Cpd1 (30 mg/kg) greatly slowed the rise in rectal temperature (Fig. 8b).

G2435R muscles have also been shown to have enhanced cation influx compared to WT which is thought to be in response to chronic reduced SR $Ca^{2+}$ load[30]. To test the effect of Cpd1 on sarcolemmal cation influx, we measured resting $Mn^{2+}$ quench of fura-2 fluorescence in isolated G2435R and WT skeletal muscle fibers. The rate of $Mn^{2+}$ quench in G2435R fibers was twice that in WT fibers (Fig. 8c, d). Cpd1 reduced $Mn^{2+}$ quench in G2435R fibers in a dose-dependent manner and the $Mn^{2+}$ quench rate at the 10 μM dose was the same as that in untreated WT fibers.

**Discussion**
Here we examined the therapeutic effects of Cpd1, an oxolinic acid-derivative RyR1-selective inhibitor with high potency[26], on the volatile anesthetic and increased environmental temperature-induced MH crisis. Cpd1 showed good solubility in normal saline (Table 1). Using three different model mice, we demonstrated

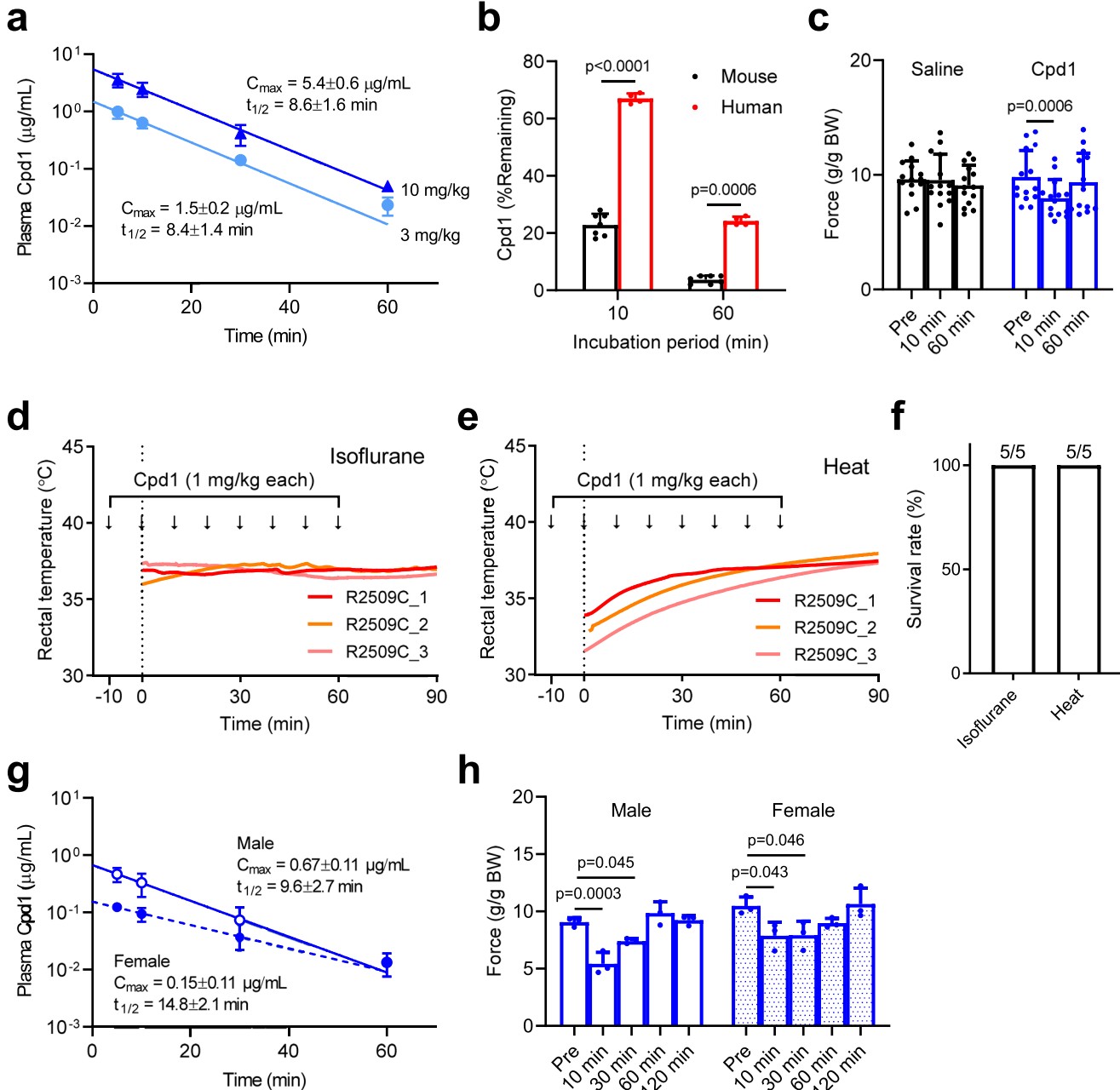

**Fig. 6 Pharmacokinetics of Cpd1 in mice. a** Averaged plasma concentration–time profiles of Cpd1 following i.p. injection of 3 mg/kg and 10 mg/kg. Data are means ± SD ($n = 6$ per time point in each group). **b** Metabolic stability of Cpd1 in the presence of mouse or human liver microsomes. Residual rates of Cpd1 at 0 min were considered 100%. Data are means ± SD (mouse: $n = 7$; human; $n = 4$) and were analyzed by two-tailed unpaired $t$-test. **c** Effect of Cpd1 on muscle force of WT mice in vivo. Muscle force was measured by grip force test (4-grips test). The tests were performed before (pre) and 10 and 60 min after i.p. injection of saline ($n = 14$) or Cpd1 ($n = 14$). The force values were normalized by body weight (BW). Data are shown as means ± SD and were analyzed by one-way ANOVA with Tukey's test. **d**–**f** Effect of repeated administration of Cpd1 on rectal temperature of R2509C mice stimulated by isoflurane (**d**) or environmental heat stress (**e**). Cpd1 was i.p. injected 8 times at 1 mg/kg every 10 min and stimulus (isoflurane or environmental heat) was started at time 0. **f** Survival rate of mice 90 min after isoflurane or heat stress challenge. All the mice survived. **g** Averaged plasma concentration–time profiles of Cpd1 following 8-times injection of 1 mg/kg at 10 min intervals in male and female mice. Data are shown as means ± SD ($n = 3$ per time point). **h** Effect of Cpd1 on muscle force of male ($n = 3$) and female ($n = 3$) WT mice in vivo following 8-times injection of 1 mg/kg at 10 min intervals. Data are shown as means ± SD and were analyzed by one-way ANOVA with Tukey's test. Source data are provided as a Source Data file.

that Cpd1: (1) prevents increase in $[Ca^{2+}]_i$ by volatile anesthetics (halothane and isoflurane) and decreases resting $[Ca^{2+}]_i$ in skeletal muscle in vitro and in vivo (Figs. 2 and 7), (2) inhibits sarcolemmal cation entry caused by chronic store $Ca^{2+}$ depletion (Fig. 8), (3) inhibits caffeine- and heat-induced contracture in isolated skeletal muscle (Fig. 3), (4) effectively prevents and treats isoflurane-induced fulminant MH crisis (Figs. 4 and 7), and (5)

treats severe heat stroke caused by environmental heat stress (Figs. 5 and 8). These results provide crucial evidence that Cpd1 is capable of effectively preventing and treating the MH crisis and heat stroke by inhibiting the RyR1 channel.

Several strains of MH mouse models carrying mutations in the *RYR1* gene have been reported to date[29,30,33,34]. We created a mouse model carrying the *RYR1*-p.R2509C mutation, which corresponds to

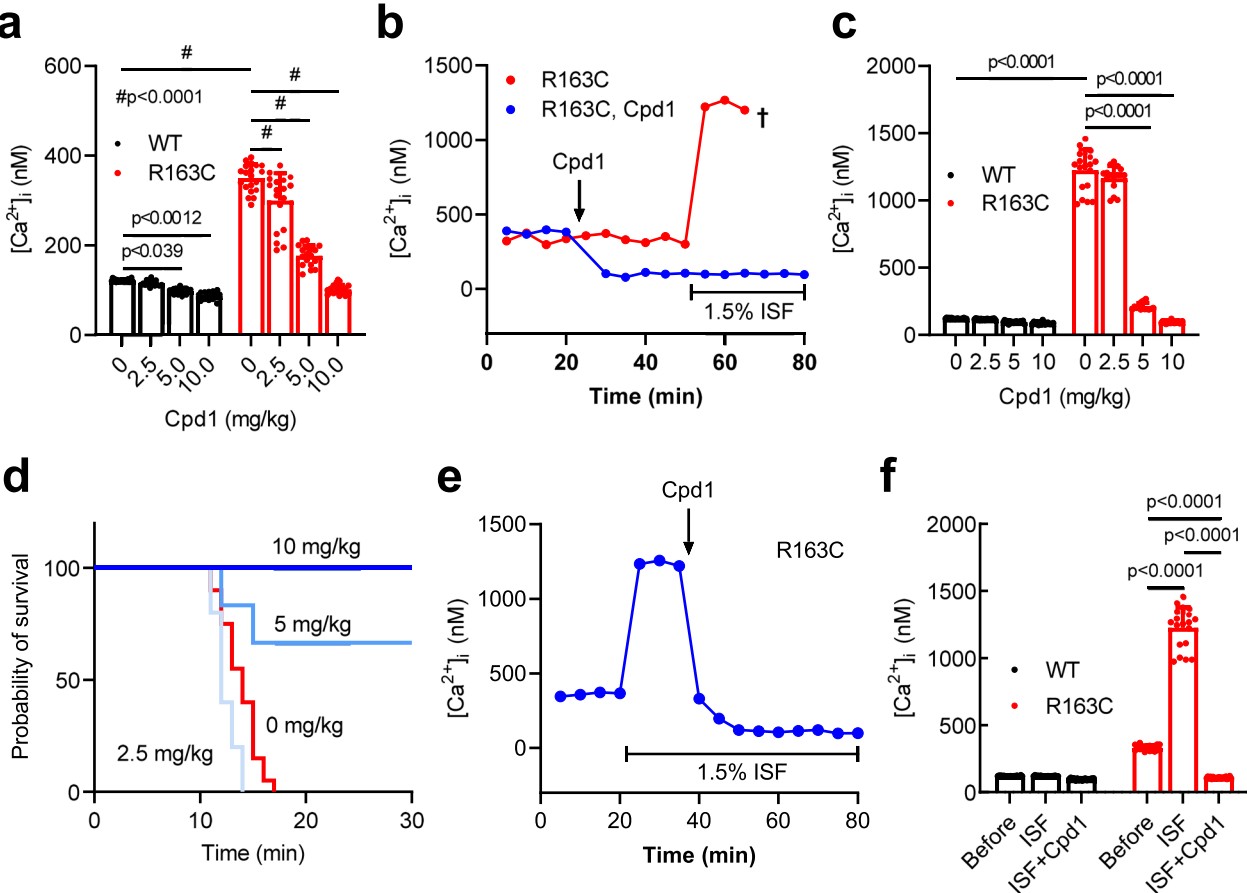

**Fig. 7 Effect of Cpd1 on heterozygous RYR1-p.R163C (R163C) mice.** Resting $[Ca^{2+}]_i$ in *vastus lateralis* muscle from R163C mice was determined in vivo by $Ca^{2+}$ selective microelectrode. **a–d** Preventive effect of Cpd1. **a** Effect of Cpd1 on resting $[Ca^{2+}]_i$ in WT and R163C mice with no stimulus. Data are shown as means ± SD (0 mg/kg: $n = 20$; 2.5 mg/kg: $n = 16$; 5 mg/kg: $n = 19$; 10 mg/kg: $n = 16$ for WT and 0 mg/kg: $n = 20$; 2.5 mg/kg: $n = 18$; 5 mg/kg: $n = 18$; 10 mg/kg: $n = 19$ for R163C) and were analyzed by two-way ANOVA with Tukey's test. **b** Time course of $[Ca^{2+}]_i$ in R163C mice during isoflurane challenge. Cpd1 (10 mg/kg) was injected i.p. at 20 min and isoflurane was administered at 50 min. **c** Effect of Cpd1 on resting $[Ca^{2+}]_i$ during 1.5% isoflurane anesthesia. Data are shown as means ± SD (0 mg/kg: $n = 18$; 2.5 mg/kg: $n = 17$; 5 mg/kg: $n = 17$; 10 mg/kg: $n = 15$ for WT and 0 mg/kg: $n = 19$; 2.5 mg/kg: $n = 15$; 5 mg/kg: $n = 16$; 10 mg/kg: $n = 21$ for R163C) and were analyzed by two-way ANOVA with Tukey's test. Cpd1 dose-dependently reduced $[Ca^{2+}]_i$ in R163C mice to a level in WT mice. **d** Survival curves of R163C mice after isoflurane anesthesia (0 mg/kg: $n = 20$; 2.5 mg/kg: $n = 5$; 5 mg/kg: $n = 17$; 10 mg/kg: $n = 9$). Cpd1 dose-dependently increases the survival rate. **e–f** Rescue effect of Cpd1. **e** Resting $[Ca^{2+}]_i$ of R163C muscle was determined by $Ca^{2+}$ selective microelectrode. Isoflurane was administered at 20 min and Cpd1 (10 mg/kg) was i.p. injected at 35 min. **f** Summary of resting $[Ca^{2+}]_i$ before (Before) and after exposure to isoflurane (ISF) and after addition of Cpd1 (ISF + Cpd1) in WT and R163C muscles. Cpd1 effectively reduces $[Ca^{2+}]_i$ in R163C muscles to the level in WT muscles. Data are shown as means ± SD (Before: $n = 20$; ISF: $n = 18$; ISF + Cpd1: $n = 16$ for WT and Before: $n = 17$; ISF: $n = 19$; ISF + Cpd1: $n = 20$ for R163C) and were analyzed by two-way ANOVA with Tukey's test. Source data are provided as a Source Data file.

the human *RYR1*-p.R2508C MH mutation[27]. In a HEK293 expression system, the R2508C mutant exhibited the highest CICR activity among mutants in the central region tested[28]. Whereas homozygous mice were embryonic lethal, heterozygous R2509C mice grew normally and were as fertile as WT (Fig. 1). R2509C mice showed a halothane- and isoflurane-induced increase in $[Ca^{2+}]_i$ (Fig. 2) and caffeine- and heat-induced contracture in skeletal muscle (Fig. 3). In addition, they exhibited a MH crisis when exposed to isoflurane anesthesia (Fig. 1) and heat stroke when exposed to environmental heat stress (Fig. 5, Supplementary Fig. 3). These properties are common to those seen in the other model mice carrying severe MH mutations in the *RYR1* gene, e.g., -p.R163C[29] and -p.Y524S[33]. Thus, R2509C mice are a useful model for MH research. Interestingly, time to death by isoflurane anesthesia of R2509C mice (64 ± 30 min, Fig. 1) was much longer than that of R163C mice (<20 min, Fig. 7). Correspondingly, increase in resting $[Ca^{2+}]_i$ appeared smaller in R2509C muscle (Fig. 2) than in R163C muscle (Fig. 7, Supplementary Fig. 8). These findings suggest that R2509C mice show milder phenotype than R163C mice.

Although dantrolene is the only compound that is available and used to treat and prevent MH, a major problem of dantrolene in clinical use is its poor aqueous solubility[22]. First, dantrolene must be solubilized with sterile water and injected or infused by the intravenous route, since it is virtually insoluble in normal saline. Second, dantrolene is prepared at a diluted concentration (0.33 mg/mL) due to its low water solubility, which makes rapid preparation difficult in emergency situations. Whereas the second problem has recently been solved by nanocrystalline suspension of dantrolene (Ryanodex®) which can be injected at a concentration of 50 mg/mL[43], the first problem remains unsolved to date. We found that Cpd1 exhibits good solubility (849 ± 58 μg/mL) in normal saline, which is 34-fold higher compared to dantrolene (25.7 ± 2.1 μg/mL) (Table 1). This enables continuous administration of the drug by infusion using the intravenous route and expands usage of the drug before and after surgery for prevention of MH and postoperative MH[44].

Another drawback of dantrolene in clinical use is its long plasma half-life (10–12 h in humans after intravenous

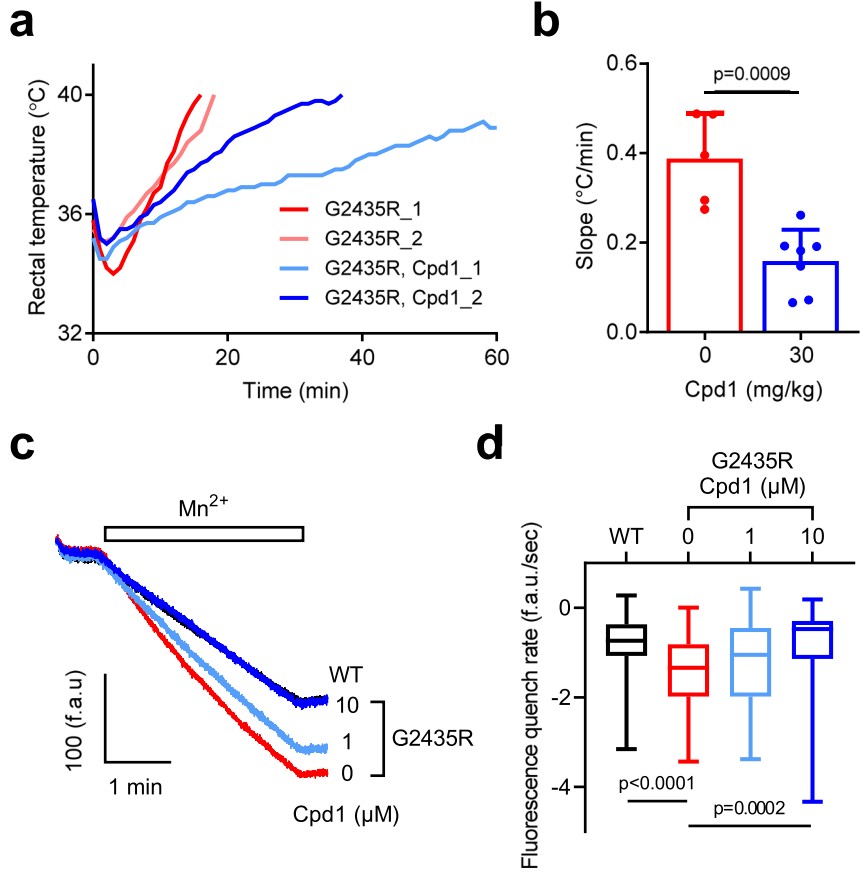

**Fig. 8 Effect of Cpd1 on homozygous RYR1-p.G2435R (G2435R) mice. a–b** In vivo heat challenge of G2435R mice. Mice were placed in a 38 °C test chamber. Cpd1 (0 or 30 mg/kg) was administered i.p. 10 min before heat challenge. **a** Time course of rectal temperature of mice. The steepest and least steep representative rectal temperature changes from 0 mg/kg and 30 mg/kg Cpd1 groups are shown. **b** Slope of temperature change from onset to endpoint at 40 °C. Data are means ± SD (0 mg/kg: $n = 7$; 30 mg/kg: $n = 5$) and were analyzed by two-tailed unpaired $t$-test. **c–d** Divalent cation entry was measured by $Mn^{2+}$ quench assay with isolated FDB muscle fibers. **c** Change in fura-2 fluorescence by switching to 1 mM $Mn^{2+}$ containing external solution. Fura-2 fluorescence was expressed as fluorescence arbitrary units (f.a.u.). The trace for G2435R with 10 μM Cpd1 almost overlapped with that for WT. **d** Slope of the decline in fura-2 fluorescence (f.a.u./sec) after switching to 1 mM $Mn^{2+}$ containing external solution. G2435R fibers show faster fluorescence quenching than WT fibers, an indicative of enhanced divalent cation entry. Cpd1 reduces the rate of fluorescence quenching in G2435R fibers in a dose-dependent manner to the level seen in WT fibers. Data are shown as box-whisker plot, with the median for all subjects shown as the center line, the box representing the 25–75 percentile, and the lines showing the range of the data (WT: $n = 78$; G2435R, 0 μM: $n = 73$; G2435R, 1 μM: $n = 39$; G2435R, 10 μM: $n = 60$) and were analyzed by one-way ANOVA with Tukey's test. Source data are provided as a Source Data file.

injection)[45]. Therefore, it is difficult to control the appropriate plasma concentration and reduce side effects such as muscle weakness[22]. Pharmacokinetic analysis revealed that Cpd1 is rapidly metabolized in mice with plasma half-life of ~8 min, probably by metabolism by the liver (Fig. 6). Muscle weakness was completely recovered within 60 min (Fig. 6). Rapid metabolism of Cpd1 was also true even after repeated administration of 1 mg/kg Cpd1 at 10 min intervals, which successfully prevented rise in the rectal temperature by exposure either to isoflurane or heat stress (Fig. 6). Thus, Cpd1 may be beneficial for a treatment of MH episodes without the concern of residual weakness.

Heat stroke is a life-threatening condition clinically diagnosed as a severe elevation in body temperature with central nervous system dysfunction[12,13]. It has been reported that MH mutations in the *RYR1* gene is implicated in some heat stroke[14–16]. Dantrolene has therapeutic effects on some patients with severe exertional heat stroke[46]. Cpd1 effectively prevents and treats fulminant heat stroke caused by environmental heat stress in both R2509C (Figs. 5 and 6) and G2435R (Fig. 8) model mice. Since Cpd1 is soluble in normal saline, it has the potential to be administered by continuous infusion which could be necessary for treatment of severe heat stroke without the concern for prolonged muscle weakness.

Dantrolene is also used for the treatment of neuroleptic malignant syndrome, a life-threatening neurologic emergency associated with the use of antipsychotic agents[47] as well as overdose of 2,4-dinitrophenol (a prohibited weight loss agent that interrupts ATP synthesis and causes hyperthermia)[48]. Since Cpd1 has similar effects in preventing and treating MH and heat stroke, it might also be a potential candidate for treatment of these emergency situations.

## Methods
**Materials.** Cpd1 was synthesized according to our reported procedure[26]. Briefly, a solution of benzo[*d*][1,3]dioxol-5-amine and diethyl ethoxymethylenemalonate in ethanol was heated at 90 °C for 18 h. After the evaporation, the residue was extracted with ethyl acetate and water. The organic layer was washed with brine, dried over sodium sulfate and concentrated under reduced pressure. The residue was purified by silica gel column chromatography to give diethyl 2-((benzo[*d*][1,3] dioxol-5-ylamino)methylene)malonate. A solution of 2-((benzo[*d*][1,3]dioxol-5-ylamino)methylene)malonate in diphenylether was heated at 250 °C for 10 h. After cooling, precipitates were corrected, washed with ether, and dried under reduced pressure to give ethyl 8-oxo-5,8-dihydro-[1,3]dioxolo[4,5-*g*]quinoline-7-carboxylate. A solution of ethyl 8-oxo-5,8-dihydro-[1,3]dioxolo[4,5-*g*]quinoline-7-carboxylate in dry DMF was added to a suspension of sodium hydride in dry dimethylformamide at 0 °C. After stirring for 10 min at room temperature, a solution of *n*-octyl bromide was added to the mixture. The reaction mixture was

heated at 55 °C for 9 h. After removal of the solvent in vacuo, the residue was extracted with ethyl acetate and water. The organic layer was washed with brine, dried over sodium sulfate, and evaporated. The residue was purified by silica gel column chromatography to give a mixture of ethyl and $n$-octyl ester of 5-octyl-8-oxo-5,8-dihydro-[1,3]dioxolo[4,5-$g$]quinoline-7-carboxylic acid, which was used in the next hydrolysis step without further separation. 2 M NaOH was added to a solution of a mixture of ethyl and $n$-octyl ester of 5-octyl-8-oxo-5,8-dihydro-[1,3]dioxolo[4,5-$g$]quinoline-7-carboxylic acid in MeOH, and the mixture was stirred at room temperature for 8 h. The reaction mixture was quenched with ice-cold 2 M hydrochloric acid and stirred for 1 h to form colorless precipitates. The precipitates were collected, washed with water, and recrystallized to give Cpd1. Sodium salt of Cpd1 was prepared as follows. Cpd1 (100 mg, 0.29 mmol) was dissolved in 3 mL of tetrahydrofuran. Then, 44 μL of NaOH aqueous solution (25%, 0.28 mmol) was added to the solution, and the suspension was sonicated for 10 min to afford white precipitate. After 24 h at room temperature, the colorless precipitates were collected by filtration, and washed with cold tetrahydrofuran. The residual material was dried under reduced pressure to give the sodium salt of Cpd1 as colorless powder.

**Determination of Cpd1 thermodynamic solubility**. Test compound (2 mg) was suspended with 1 mL of normal saline, pure water, or 5% glucose aqueous solution in glass tube, and the suspension was shaken at 150 rpm for 48 h at 25 °C. An aliquot was filtered by PVDF filter unit (Mini-Uni Prep™, GE Healthcare), and the filtrate was diluted with pure water. The diluted sample solution was injected into HPLC system (column (Mightysil RP-18, Kanto Chemical Co. Inc.), UV/vis detector (UV-2077, JASCO), pump (PU-2089, JASCO), and column oven (CO-965, JASCO)) and the peak area was recorded at 254 nm. The concentration of solution was calculated with a calibration curve. Examinations of all conditions were performed three or four times.

**Animals**. All animal-handling procedures were in accordance with the guidelines and approved by the ethics committees of the Jikei University School of Medicine, Juntendo University School of Medicine, the National Center of Neurology and Psychiatry (NCNP), Mount Sinai Hospital and UK Home Office, administered by the University of Leeds. Mice were housed in isolator cages, fed with food and water ad libitum, and kept under controlled environment with 12/12 light/dark cycles, 23–25 °C temperature, and 50–60% relative humidity in specific pathogen-free conditions in the Jikei University, Juntendo University, National Institute of Neuroscience, NCNP, Mount Sinai Hospital and the Leeds Institute of Medicine animal facilities.

**Generation of *RYR1*-p.R2509C mouse using the CRISPR-Cas9 gene editing system**. Mouse genomic sequence within upstream- and downstream-50 bps each to that corresponding to human mutation site was searched by CRISPR design tool (http://crispr.mit/edu) to select PAM and its consequent guide sequence with high specificity. CAATGCCATACACTCGGTCCAGG was selected for crRNA synthesis. A single-stranded donor oligonucleotide carrying Xba I site, CCGAAGATGTCTGCATCCTTCGTGCCTGACCAGGCATCCATGGTGCTTTTTCTAGACTGCGTGTATGGCATTGAGAACCAGGACTTCTTGCTGCATGTGCTGGATGTGGG was chemically synthesized for homologous recombination (Fig. 1). A *RYR1*-genomic fragment containing PAM and the consequent guide sequence were amplified by PCR from wild-type mouse genomic DNA. RNP complex formed by mixing of tracrRNA and crRNA with recombinant Cas9 protein, was added to the PCR fragment and incubated for 1 h. Cleavage of PCR fragment into two fragments was confirmed by agarose electrophoresis. RNP complex and single-stranded donor oligonucleotide were introduced to mouse one-cell-stage zygotes by electroporation. The electroporated zygotes were transferred to the oviducts of pseudopregnant females. Genomic DNA was prepared from newborns' tail and knock-in mice were screened by PCR-RFLP method using a pair of primers (Supplementary Table 1) and Xba I digestion (Fig. 1). Crossing obtained knock-in founders with wild-type mice to get F1 generations were done by standard protocol.

**Preparation of single flexor digitorum brevis (FDB) cells**. WT and mutant mice (R2509C, 8- to 20-week old; R163C, 12- to 16-week old, and G2435R, 12- to 16-week old) were euthanized and flexor digitorum brevis (FDB) muscles were dissected and incubated with 2 mg/mL collagenase (Worthington Biochemical Co., NJ, USA) in the HEPES-Krebs solution containing 2 mg/mL bovine serum albumin for 2–3 h at 37 °C. Following incubation, single cells were separated by gentle trituration in collagenase-free HEPES-Krebs solution. Isolated single FDB cells were seeded on a laminin-coated cover slip of the 35 mm glass-bottom dish or on 96 well plate coated with Cultrex basement membrane matrix (Trevigen, Minneapolis, USA).

**Ca$^{2+}$ imaging of isolated single FDB cells**. Isolated single FDB cells were incubated with 4 μM Cal520-AM (AAT Bioquest, CA, USA) or fura-2-AM (Thermo Fisher Scientific) dye in the HEPES-Krebs solution (140 mM NaCl, 5 mM KCl, 2 mM CaCl$_2$, 1 mM MgCl$_2$, 11 mM glucose, 5 mM HEPES, pH 7.4) for 30 min at 37 °C. The cells were then washed three times with the HEPES-Krebs solution to remove excess dye, and after 30 min of de-esterification, fluorescence images were obtained with a 20X (NA = 0.75) objective lens of an inverted microscope (Nikon TE2000-E Japan) equipped with the sCMOS camera (Zyla, Andor, Belfast, Northern Ireland) using Metamorph v7 (Molecular Devices). Cal520 was excited at

480 ± 15 nm, and fluorescence was measured at wavelengths of 525 ± 25 nm. Fura-2 was excited at alternating lights of 340 ± 6 nm and 380 ± 6 nm, fluorescent light was detected at 510 ± 40 nm, and the ratio of fluorescence excited at 340 nm to that at 380 nm was determined. Imaging experiments were carried out at 26 °C by superfusing a HEPES-Krebs solution with or without halothane or isoflurane. Cpd1 was administered 5 min before measurements. For Mn$^{2+}$ quenching experiments, fura-2 was excited at 360 ± 12 nm and emitted fluorescence at 510 ± 40 nm was monitored. The cells were perfused with basal Krebs solution (140 mM NaCl, 5 mM KCl, 1 mM MgCl$_2$, 11 mM glucose, 2 mM CaCl$_2$, 10 mM HEPES, pH 7.4) for 60 s to obtain the baseline signal and then perfused with Mn$^{2+}$ quenching Krebs solution (140 mM NaCl, 5 mM KCl, 1 mM MgCl$_2$, 11 mM glucose, 0.5 mM MnCl$_2$, 10 mM HEPES, pH 7.4) for 180 s followed by reperfusion with basal Krebs solution for 120 s.

**Measurement of muscle force in vitro**. Muscle contraction measurement was assessed using intact soleus muscle as follows[49]. WT and R2509C mice (8- to 20-week old) were anesthetized with an i.p. injection of sodium pentobarbital (70 mg/kg body weight) and the soleus muscles were dissected. The isolated muscles were mounted between a force transducer (UL-100; Minebea Co., Tokyo, Japan) and a fixed hook in a chamber containing Krebs solution (140 mmol/L NaCl, 5 mmol/L KCl, 2 mmol/L CaCl$_2$, 1 mmol/L MgCl$_2$, 1 mmol/L NaH$_2$PO$_4$, 25 mmol/L NaHCO$_3$, and 11 mmol/L glucose) bubbled with 95% O$_2$ and 5% CO$_2$ at 25 °C. The muscle force was recorded and analyzed by LabChart v6 (AdInstruments, Dunedin, New Zealand). The muscles were stretched to determine the length which produced maximum twitch contraction in response to field stimulation for 1 msec at supramaximal voltage. For caffeine contracture studies, muscles were exposed for 2 min to Krebs solution containing 10, 15, or 20 mM caffeine. For heat-induced contracture studies, muscles were exposed to Krebs solution warmed at 27, 35, 40, and 42 °C until basal force peaked. For twitch and tetanic contraction measurements, muscles were stimulated at 1 min intervals with 20 trains of 1, 2, 5, 10, 20, 50, 100, and 200 Hz pulses. For Cpd1 treatment, muscles were incubated with Krebs solution containing 3 μM Cpd1 15 min prior to measurements. Absolute contractile force was normalized to the maximum diameter of the muscle.

**In vivo isoflurane challenge**. WT and R2509C mice (8- to 20-week old) were weighed and then placed into an anesthetic chamber. Anesthesia was induced with 2.0–3.0 vol% isoflurane in air using a precision vaporizer (TK-7w BioMachinery isoflurane system, Japan) until there was no detectable response to tweezer pinches (within 60 s). Then, mice were rapidly placed atop a bed prewarmed at 37 °C and anesthesia was maintained with isoflurane, 1.5–2.0 vol% in air via a nose cone attachment. Rectal temperature was continuously monitored with a rectal probe (RET-3; Physitemp Instruments, LLC, NJ, USA) using Powerlab 26 T (AdInstruments, Dunedin, New Zealand). Data were recorded every 10 s for up to 90 min or until the animal died of an MH crisis. Cpd1 was administered i.p. either 10 min before the start of isoflurane anesthesia for test of the preventive effect or when the rectal temperature was achieved 39 °C for treatment of MH episodes.

**In vivo heat stress challenge**. In anesthetized R2509C mice: to investigate heat stress responses, WT and R2509C mice (8- to 20-week old) were anesthetized i.p. with an anesthetic mixture (0.75 mg/kg medetomidine, 4 mg/kg midazolam, and 5 mg/kg butorphanol) and transferred to a test chamber prewarmed at 35 °C. Rectal temperature was measured continuously during the heat stress challenge (up to 120 min) or until death of animals by fulminant heat stroke. Cpd1 was administered i.p. either 10 min before the start of heat stress challenge for test of the preventive effect or when the rectal temperature was achieved 38 or 39 °C for treatment of heat stroke.

In awake restrained G2435R mice: in this model to investigate the heat stress response, G2435R mice (12- to 16-week old) was administered Cpd1 i.p., placed into a mouse restrainer (Kent Scientific, CT, USA), and placed in an environmental chamber warmed to 38 °C. Rectal temperature was monitored continuously with a probe and recorded every min. When the rectal temperature reached 40 °C the experiment was ended and the animal was euthanized by cervical dislocation.

**Ca$^{2+}$ determinations in intact muscle fibers using ion-specific microelectrodes**. Ca$^{2+}$ ion specific microelectrode recordings were performed as follows[38,50]. For in vivo measurements, WT and heterozygous R163C mice (12- to 16-week old) were anesthetized by i.p. injection of 100 mg/kg ketamine and 5 mg/kg xylazine and placed on a 37 °C heating pad and their temperature was monitored. The *vastus lateralis* was exposed surgically after which its muscle fibers were impaled with double-barreled Ca$^{2+}$-selective microelectrode, and the potentials were recorded via a high-impedance amplifier (WPI Duo 773 electrometer; WPI, Sarasota, FL, USA). The potential from the 3 M KCl microelectrode ($V_m$) was subtracted electronically from the potential of the Ca$^{2+}$ electrode ($V_{CaE}$) to produce a differential Ca$^{2+}$-specific potential ($V_{Ca}$) that represents the [Ca$^{2+}$]$_i$. $V_m$ and $V_{Ca}$ were filtered (30–50 kHz) to improve the signal-to-noise ratio and stored in a computer for further analysis. Determinations were carried out at resting condition and then 5–10 min after commencing exposure to 1.5% isoflurane in the inspired air. Cpd1 (0–10 mg/kg) was administered i.p. before exposure to isoflurane and measurements made every 5 min. For in vitro measurements, single flexor

digitorum brevis (FDB) muscle fibers were enzymatically isolated as described above and placed in mammalian Ringer solution of following composition (in mM): 140 NaCl, 5 KCl, 2 CaCl$_2$, 1 MgCl$_2$, 5 glucose, and 10 HEPES, pH 7.4. Muscle fibers were impaled with the microelectrodes as above and [Ca$^{2+}$]$_i$ was determined in the presence of varying (0–5 µM) concentrations of Cpd1.

**Mn$^{2+}$ quench assay**. Assays were carried out as follows[38]: 5 µM of fura2-AM dye (Thermo Fisher Scientific) dissolved in mammalian Ringer solution (133 mM NaCl, 5 mM KCl, 1 mM MgSO$_4$, 25 mM HEPES, 5.5 mM glucose, 2 mM CaCl$_2$, pH 7.4) was loaded into dissociated WT and homozygous G2435R FDB fibers in 96 well plates for 30 min at 37 °C, 5% CO$_2$. Fibers were then washed twice with Ringer solution and then incubated with the desired concentration of Cpd1 dissolved in Ringer solution for the final wash in the drug-treated samples. After washing fibers were maintained at 37 °C for 20 min for de-esterification of fura2-AM and in the presence of the tested concentration of Cpd1. The fibers were then transferred to the stage of a Nikon TE2000 inverted microscope (Nikon, Tokyo, Japan). Fura-2 was illuminated at its isosbestic wavelength (360 nm) with the X-Cite 120 metal halide light source (Exfo, Ontario, Canada) and fluorescence emission at 510 nm was recorded using a 10 ×0.3na objective. The fibers were perfused with Ringer solution for 30 s to obtain the baseline signal and then perfused with manganese quench buffer (133 mM NaCl, 5 mM KCl, 0.5 mM MnCl$_2$, 25 mM HEPES, 5.5 mM glucose, pH 7.4) for 180 s followed by reperfusion with Ringer solution for 30 s to wash away the manganese quench buffer. Data were collected using an intensified 10-bit digital intensified CCD at 30 fps (Stanford Photonics, Stanford, CA) from regions of interest on 3–6 individual fibers per well and analyzed using Piper software v1.3.04 (Stanford Photonics, Stanford, CA). The fluorescence signals were plotted as fluorescence arbitrary units (f.a.u.) over time. Prism v7 (GraphPad Software, Inc., La Jolla, USA) was used to fit the baseline signal and quench signal into linear regression models. The fluorescence quench rate was calculated by subtracting the slope of the baseline signal from that of the quench signal to obtain the net slope for both drug-treated and untreated samples.

**Pharmacokinetics**. Cpd1 sodium salt (3 mg/kg and 10 mg/kg) dissolved in normal saline was administered i.p. to WT C57BL/6 J mice (10 week-old). Serial tail bled blood samples (∼20 µL) were collected using heparinized tip at 5, 10, 30, and 60 min after administration. Aliquots (5 µL) of plasma samples obtained from each blood were treated with 50 µL of acetonitrile and organic layer were injected onto LC–MS/MS system (Waters Corp., Milford, MA, USA). The pharmacokinetic parameters included maximum concentration (C$_{max}$), elimination half-life (T$_{½}$).

**Metabolic stability in liver microsomes**. Cpd1 (final concentration was 0.5 µM) was incubated with human or mouse hepatic microsomes (XenoTech Ltd, Kansas City, KS) with nicotinamide adenine dinucleotide phosphate (NADPH) for 10 and 60 min at 37 °C. Each reaction was stopped by acetonitrile, and the value of unchanged compound 1 was determined by quantitating using LC–MS/MS, and the metabolic stability was evaluated as a remaining rate (%).

**Measurement of muscle force in vivo**. Muscle force was measured in vivo by grip-force test. The tests were performed as blinded, so the experimenter did not know which kind of solution (vehicle or Cpd1) was injected to each mouse. The force was measured before and 10 and 60 min after i.p. injection of saline or Cpd1 (10 mg/kg). Mice (8- to 12-week old) were gently held by the tails and allowed them to grasp the horizontally positioned metal bar of the Grip Strength Meter (MK308M, Muromachi Kikai, Tokyo, Japan) with their forelimbs and hindlimbs (4-grips test). Each mouse performed four trials for force measurement at every time point. The highest force value applied to the metal bar was recorded and normalized by body weight[51].

**Statistics**. Data are presented as means ± SD. Unpaired Student's *t* test was used for comparisons between two groups. One-way or two-way analysis of variance (ANOVA), followed by Tukey's test, was performed to compare three or more groups. Two-tailed tests were used for all analyses. Statistical analysis was performed using Prism v7 and v8 (GraphPad Software, Inc., La Jolla, USA).

**Reporting summary**. Further information on research design is available in the Nature Research Reporting Summary linked to this article.

## Data availability

All data generated or analyzed in this study are available within the article and its Supplementary Information. All raw data supporting the findings from this study are available from the corresponding author upon request. Source data are provided with this paper.

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

## Acknowledgements

We are grateful for Mariko Hanano (The Jikei University), Ikue Hiraga (Juntendo University) and the Laboratories of Molecular and Biochemical Research, Research Support Center (Juntendo University), for their technical support. This work was supported in part by the Japan Society for the Promotion of Sciences KAKENHI (grant Number 19K07306, 19H03198 and 20H04511 to T.Y., 20K11368 to T.K., 19K07105 to N. K., 19K06955 to S.K., and 19H03404 to T.M.), the USA National Institute of Arthritis, Musculoskeletal and Skin Diseases (R01AR068897; P.D.A., J.R.L., C.D. and X.L.), the Platform Project for Supporting Drug Discovery and Life Science Research (Basis for Supporting Innovative Drug Discovery and Life Science Research (BINDS)) (JP20am0101098 to H.K. (support number 1858), JP20am0101085 to K.N., JP20am0101123 to S.N. (support number 1424), and JP20am0101080 to T.M.), the Practical Research Project for Rare/Intractable Diseases (19ek0109202 to N.K.) from the Japan Agency for Medical Research and Development (AMED), and an Intramural Research Grant (29-4 and 2-5 to I.N. and T.M., 30-9 and 2-6 to S.Noguchi) for Neurological and Psychiatric Disorders of NCNP, the Vehicle Racing Commemorative Foundation (6114 and 6237) to T.M, and Cooperative Research Project of the Research Center for Biomedical Engineering to H.K.

## Author contributions

T.Y., T.K., N.K., and T.M. conceived and designed the work. T.Y., T.K., N.K., M.K., S.Noguchi, T.I., Y.U.I., S.M., H.I., N.M., A.U., J.A., J.R.L, X.L., C.D., S.K., K.I., B.L., Y.I., K.N., and T.M. acquired and analyzed data. T.Y., T.K., N.K., M.K., S.Noguchi, T.I., Y.U.I., I.N., S.M., H.I., N.M., H.K., A.U., J.A., J.R.L., X.L., C.D., P.D.A., S.K., K.I., B.L., Y.I., K.N., S.N, T.S., and T.M. interpreted data. T.Y., T.K., N.K., S.K., J.R.L, P.D.A., and T.M. wrote and revised the manuscript. All authors reviewed and approved the revised manuscript.

## Competing interests

The authors declare no competing interests.
