## [Peer Review File · Nature Communications]

Reviewers' Comments:

Reviewer #1:

Remarks to the Author:

Comments to authors:

Malignant hyperthermia is a hypermetabolic disorder which manifests when patients are exposed to pharmacological triggering agents. The disorder is potentially fatal if untreated. The only approved treatment for MH is dantrolene which has several limitations to its use. In this manuscript the investigators report on the efficacy of a novel RyR1-selective inhibitor in treating MH attacks in a newly developed mouse model of the disease. The authors find here that the drug is effective in treating and rescuing MH in the R2509C mouse and was able to prevent halothane-induced increases in cytoplasmic Ca²⁺ and partially prevented caffeine-contractures in mutant muscle. The drug shows promise in treating a range of MH-linked RYR1 mutations since it could normalize Ca²⁺ handling in muscle fibers from two other established mouse models of MH. Compound 1 was unable to prevent heat stroke due to environmental heat exposure but there was some evidence it could treat mice already undergoing heat stroke. The authors also find that Compound 1 is superior to dantrolene with regard to solubility and pharmacokinetics. While the results show that Compound 1 is a promising therapeutic option for MH, the evidence for its efficacy as a treatment for heat stroke is unconvincing in its current form. This manuscript is likely to be of interest to the malignant hyperthermia and muscle research communities interested in the development of RyR1 inhibitory compounds but there are several issues that need to be addressed:

MAJOR ISSUES

1. In the abstract it is stated that Compound 1 effectively rescues MH and heat stroke in several mouse models relevant to MH, however there is only evidence for this effect in the R2509C mice. The ability of the drug to normalize cytoplasmic Ca²⁺ (R163C) and decrease sarcolemmal Ca²⁺ influx in the G2435R model is not sufficient evidence that Compound 1 can also treat MH in those models. The idea of testing Compound 1 on multiple MH models provides strong evidence for the efficacy of the drug, but this needs to be tested more thoroughly
2. Functional analyses from the literature show that different mutations have variable effects on RyR1 and may result in different effects on cytoplasmic Ca²⁺, SR load etc. Since this mouse model has not been characterized before it should be shown whether it has increased resting cytoplasmic Ca²⁺ (like the R163C muscle) or increased sarcolemmal Ca²⁺ influx as in the G2435R muscle, rather than extrapolating the results obtained in those two mouse models to the new mouse model.
3. In the discussion it is proposed that continuous IV infusion of Compound 1 may be necessary due to the rapid clearance and transient effect of the drug during MH episodes. Has plasma clearance, liver metabolism or muscle function been analyzed following longer term IV administration or after repeated IP injections? These studies would be valuable in assessing the safety of the drug, and should also be repeated in female mice.
4. Has this drug been tested in intact skeletal muscle before? Has there been any work done to confirm that in skeletal muscle the drug is specifically targeting RyR1? For example, have you repeated the caffeine contracture study with 4-CMC or ryanodine?
5. The grip strength tests shown in Figure 6 should be repeated in mutant mice. It should also be stated whether the investigators who did grip strength tests were blinded.
6. How long were muscles and fibers treated with Compound 1? Also, it is unclear whether the sodium salt was used for all experiments (including isolated muscles and fibers) or just the in vivo experiments. If two different forms were used it should be clearly stated which form was used for each experiment. Ideally the same form of drug that was used to prevent MH in vivo should also

be used in the in vitro experiments.

7. How long was mouse survival followed? It is only stated that mice treated with 10mg/kg Compound 1 survived for 60 min, but in patients and mouse models of MH death may occur at a later stage after drug/heat exposure. Did you ever monitor the mice for a 24hr period after isoflurane or heat exposure?

8. The data demonstrate that the R2509C model has enhanced heat sensitivity however there was no in vitro work done to show that the same cellular responses (calcium release, muscle contraction) that are triggered by isoflurane are also triggered by heat. The evidence for Compound 1 being an effective therapy for heat sensitivity in these mice is limited at the moment. While the drug does appear to delay death when given to mice already experiencing heat-induced hyperthermia, by the end of the 60 min exposure rectal temperature was clearly increasing. Therefore, these results would be more convincing if mouse survival and rectal temperature was monitored after the completion of the 60 min heat exposure.

MINOR ISSUES:

1. Figure 1 - Time to death is compared for male and female mice, but max rectal temperature is only shown for males. Was there a difference between males and females?
2. Compound 1 causes a significant decrease in soleus force production from het mice. Were there any effects of Compound 1 on WT muscle force production?
3. What age mice are used? This should be stated in the methods and/or results section.
4. The correct nomenclature should always be used when referring to the mouse strain instead of using the terms "het" and "hom", especially since multiple mouse models are used. There's also inconsistent use of "Wt" versus "WT".
5. Line 121 –provide a citation for halothane stimulation of RyR1 and its use in MH testing.
6. Figure 2 – individual data points should be shown for panel D.
7. Figure 5 – Individual data points should be shown for panel D.
8. Line 379 – change "x" to "X"
9. There are many examples of incorrect wording and grammar throughout the text. Please proofread carefully and correct so that the text is easier to read. Some examples:
 - a. Line 160 – change "inducing" to "induction of"
 - b. Line 173 – change "of existing" to "in other"
 - c. Line 176 – change "was maintained a relatively" to "was maintained at a relatively"
 - d. Line 182 – add "the" after tested
 - e. Line 188 – change "at" to "in"
 - f. Line 419 – add "at" after achieved
 - g. Line 487 – change "points" to "point".
 - h. Line 667 – change "fully" to "full"

Reviewer #2:

Remarks to the Author:

Comments for Authors: Malignant hyperthermia (MH), a life-threatening disorder triggered by administration of volatile anesthetics (halothane, isoflurane, etc.), has been linked to mutations in the type-1 ryanodine receptor (RyR1), the Ca²⁺ release channel in skeletal muscle. Similar hyperthermic reactions (known as heat stroke; HS) can be triggered by exposure to high environmental temperature.

Dantrolene is the only approved drug for acute treatment of MH crises (though, not used in case of HS). Nevertheless, dantrolene has some disadvantages for clinical use: a) poor water solubility which makes its rapid preparation difficult in emergency situations; b) long plasma half-life, which causes side effects in patients (such as prolonged muscle weakness).

No alternative drugs have been developed over 50 years to replace Dantrolene.

In this paper the authors created and characterized a new MH mouse model RYR1-p.R2509C; and identified in a oxolinic acid derivative (Compound 1, which was already reported as an inhibitor of RyR1 in 2019) a drug that effectively prevents and rescues MH and HS crisis triggered by isoflurane and heat in mice.

The results are important as they characterize a new drug which has the potential to replace dantrolene.

However, the paper may need some adjustments in the way data are currently presented.

Main Comments:

1. Fig. 2d and e: could the authors explain why they used Soleus muscle (a slow twitch muscle) for these IVCTs? Muscles in mice are mostly fast-twitch.
2. Figs. 2 and 3: could the authors explain why they used Halothane in Fig 2b and c, but Isoflurane in Fig. 3?
3. Comparison between Figs. 3 and 4:
 - a. compound 1 was administered in one case when rectal temperature reached 39oC, but at 38oC in the second case: can the authors explain why?
 - b. while Fig 3 is complete, it looks like Fig. 4 is INCOMPLETE, as in panels a-c the dose 3mg/kg is missing..... ??
 - c. It would be nice to have Figs.3 and 4 showing the same sets of data. Currently, panels b and c are different in the two Figures (while panel a and the 3 bottom panels are the same in both).
4. Fig. 5 shows in panel a) intracellular Ca²⁺ concentrations at rest, and in panel b) Mn²⁺ quench of fura-2 fluorescence. Nice to have those data included in the manuscript. However, they were collected in samples from MH mice different from the main knock-in mouse line used in the rest of the experiments. Why?
Could this figure be moved into supplemental, and maybe rescue Suppl. Figures with data collected in the RYR1-p.R2509C knock-in mouse?

Minor Comments:

Title:

A novel saline-soluble, rapidly-metabolized RyR1 inhibitor rescues volatile anesthesia induced death and environmental heat stroke in a mouse model relevant to malignant hyperthermia.

Title could be improved.

Suggestion 1: rescues volatile anesthesia-induced death and...

Suggestion 2: A novel RyR1 inhibitor prevents and rescues sudden death in a mouse model of malignant hyperthermia and heat stroke.

Introduction:

Page 3, line 55: Correct as follow: This MECHANISM is referred to as....

Page 3, line 56: The RyR1 channel can also be directly activated by Ca²⁺.... I was under the impression that Ca²⁺ induced Ca²⁺ release was a mechanism used primarily by RyR2 and RyR3, while RyR1 is less sensitive to it.

Results:

Page 5, line 104-105: Correct as follow: ...was similar among individuals MICE, and....

Page 5, line 114: Correct as follow: We initially tested the effect of Compound 1 IN in vitro experiments USING SINGLE FIBERS AND ISOLATED MUSCLES. Indeed, the authors in Figure 2 describe results in isolated FDB fibers and in whole Soleus muscles.

Page 5, line 124: Correct as follow: ... in het fibers, but not IN Wt....

Page 6, line 129: ... of MH susceptibility in humans (please add REF). Please provide a reference after this statement.

Materials and Methods:

Page 16, line 388 & page 18, line 443: Remove Briefly, as the sections that follow are not brief

Figures:

Figure 1e, Y axis: should it be "maximum rectal temperature" as in Fig 3b?

Figs. 3 and 4, panels a and d: a color/symbol legend (as - for example - in Figure 2d and e) would probably be better.

Compare Figs. 3e and 4e: at endpoint is missing

Point-by-point responses to reviewers' comments.

We really appreciate the positive and constructive comments made by the reviewers. According to the reviewers' comments, we have carried out a lot of additional experiments, especially therapeutic effects of Compound 1 (Cpd1) on heterozygous RYR1-p.R163C and homozygous RYR1-p.G2435R mice. Point-by-point responses to the reviewers' comments are described below.

Reviewer #1 (Remarks to the Author):

Comments to authors:

Malignant hyperthermia is a hypermetabolic disorder which manifests when patients are exposed to pharmacological triggering agents. The disorder is potentially fatal if untreated. The only approved treatment for MH is dantrolene which has several limitations to its use. In this manuscript the investigators report on the efficacy of a novel RyR1-selective inhibitor in treating MH attacks in a newly developed mouse model of the disease. The authors find here that the drug is effective in treating and rescuing MH in the R2509C mouse and was able to prevent halothane-induced increases in cytoplasmic Ca²⁺ and partially prevented caffeine-contractions in mutant muscle. The drug shows promise in treating a range of MH-linked RYR1 mutations since it could normalize Ca²⁺ handling in muscle fibers from two other established mouse models of MH. Compound 1 was unable to prevent heat stroke due to environmental heat exposure but there was some evidence it could treat mice already undergoing heat stroke. The authors also find that Compound 1 is superior to dantrolene with regard to solubility and pharmacokinetics. While the results show that Compound 1 is a promising therapeutic option for MH, the evidence for its efficacy as a treatment for heat stroke is unconvincing in its current form. This manuscript is likely to be of interest to the malignant hyperthermia and muscle research communities interested in the development of RyR1 inhibitory compounds but there are several issues that need to be addressed:

MAJOR ISSUES

1. In the abstract it is stated that Compound 1 effectively rescues MH and heat stroke in

several mouse models relevant to MH, however there is only evidence for this effect in the R2509C mice. The ability of the drug to normalize cytoplasmic Ca²⁺ (R163C) and decrease sarcolemmal Ca²⁺ influx in the G2435R model is not sufficient evidence that Compound 1 can also treat MH in those models. The idea of testing Compound 1 on multiple MH models provides strong evidence for the efficacy of the drug, but this needs to be tested more thoroughly.

We carried out additional *in vivo* experiments about the effect of Compound 1 (Cpd1) on MH model mice; isoflurane-induced MH episodes with R163C mice (**Fig. 7**) and heat stroke by environmental heat stress with G2435R mice (**Fig. 8**). We show that Cpd1 successfully prevented and treated MH episodes induced by isoflurane and delayed the onset of heat stroke. These observations were described in Results (p. 11, lines 251-265; p. 11, lines 273-278).

2. Functional analyses from the literature show that different mutations have variable effects on RyR1 and may result in different effects on cytoplasmic Ca²⁺, SR load etc. Since this mouse model has not been characterized before it should be shown whether it has increased resting cytoplasmic Ca²⁺ (like the R163C muscle) or increased sarcolemmal Ca²⁺ influx as in the G2435R muscle, rather than extrapolating the results obtained in those two mouse models to the new mouse model.

We determined resting [Ca²⁺]_i (**Fig. 2**) and sarcolemmal Mn²⁺ influx (**Supplementary Fig. 2**) of skeletal muscles from R2509C mice using fura-2. We found that [Ca²⁺]_i in R2509C muscle was significantly higher than that of WT and Cpd1 reduced it to the level of WT. Unexpectedly, Mn²⁺ influx was not different than WT in R2509C muscle. These observations were described in Results (p. 6, lines 129-136, 137-140).

3. In the discussion it is proposed that continuous IV infusion of Compound 1 may be necessary due to the rapid clearance and transient effect of the drug during MH episodes. Has plasma clearance, liver metabolism or muscle function been analyzed following longer term IV administration or after repeated IP injections? These studies would be valuable in assessing the safety of the drug, and should also be repeated in female mice.

We repeated plasma clearance, liver metabolism and muscle function of Cpd1 after single injection in female mice and found no sex differences (**Supplementary Fig. 5**). In addition, we examined plasma clearance and muscle function after repeated administrations of Cpd1 (1 mg/kg, 8 times at 10 min intervals) (**Fig. 6g**). These observations were described in Results (p. 9, lines 222-223; p. 10, lines 233-234, 242-246).

4. Has this drug been tested in intact skeletal muscle before? Has there been any work done to confirm that in skeletal muscle the drug is specifically targeting RyR1? For example, have you repeated the caffeine contracture study with 4-CmC or ryanodine?

We show in the current manuscript that Cpd1 reduces resting Ca^{2+} (**Figs. 2 and 7, Supplementary Fig. 8**) and prevents halothane- and isoflurane-induced Ca^{2+} release (**Fig. 2**) in MH susceptible skeletal muscle. These findings strongly suggest that Cpd1 inhibits RyR1. We cannot completely exclude the possibility that Cpd1 also targets other proteins. However, the fact that there are no apparent side effects of Cpd1 other than muscle weakness in mice strongly suggests that Cpd1 specifically targets RyR1 in skeletal muscle. According to the reviewer's suggestion, we carried out contracture study with 4-CmC (**figures for reviewers**). Although 4-CmC elicited muscle contracture, the effect was not different between WT and R2509C muscles. This is consistent with the previous report that 4-CmC cannot discriminate MH status (Weigl et al., *Anesth Analg* 99:103-107, 2004).

5. The grip strength tests shown in Figure 6 should be repeated in mutant mice. It should also be stated whether the investigators who did grip strength tests were blinded.

We carried out the grip strength tests in R2509C mice (**Supplementary Fig. 6**). We found that muscle weakness was also observed in 10 min but recovered within 60 min after administration of Cpd1. These observations were described in Results (p. 10, lines 233-234). We stated in Methods that the investigators who performed the tests were blinded (p. 22, line 541-543).

6. How long were muscles and fibers treated with Compound 1? Also, it is unclear whether the sodium salt was used for all experiments (including isolated muscles and fibers) or just the in vivo experiments. If two different forms were used it should be clearly stated which form was used for each experiment. Ideally the same form of drug that was used to prevent MH in vivo should also be used in the in vitro experiments.

Muscle cells were pre-treated with Cpd1 for 5-15 min depending on the experiments as stated in the Methods section (p. 18, lines 426 and 448-449). We used Cpd1 sodium salt for all the experiments. To avoid confusion, we moved the description about the use of Cpd1 sodium salt to the beginning of the results (p. 5, lines 111-119).

7. How long was mouse survival followed? It is only stated that mice treated with 10mg/kg Compound 1 survived for 60 min, but in patients and mouse models of MH death may occur at a later stage after drug/heat exposure. Did you ever monitor the mice for a 24hr period after isoflurane or heat exposure?

We monitored the mice for at least 24 h after isoflurane and heat exposure and found that all the survivors behaved normally. These observations were described in Results (p. 7, lines 173-174, p. 8, 183-184; p. 9, lines 209-211).

8. The data demonstrate that the R2509C model has enhanced heat sensitivity however there was no in vitro work done to show that the same cellular responses (calcium release, muscle contraction) that are triggered by isoflurane are also triggered by heat. The evidence for Compound 1 being an effective therapy for heat sensitivity in these mice is limited at the moment. While the drug does appear to delay death when given to mice already experiencing heat-induced hyperthermia, by the end of the 60 min exposure rectal temperature was clearly increasing. Therefore, these results would be more convincing if mouse survival and rectal temperature was monitored after the completion of the 60 min heat exposure.

To address the cellular responses triggered by heat, we carried out additional experiments of muscle contraction. We demonstrated a larger contracture of soleus muscles in R2509C muscle than in WT muscle when the bath temperature was raised to

42 °C and this was inhibited by Cpd1 (**Fig. 3c, 3d**). These observations were described in Results (p. 7, lines 151-154). Heat-induced contracture is consistent with the previous finding with Y522S MH model mice (Chelu et al., 2004). In the *in vivo* experiments all the survivors behaved normally for at least 24 h after being removed from the heat stress environment, and regaining consciousness. These observations were described in Results (p. 9, lines 209-211).

MINOR ISSUES:

1. Figure 1 - Time to death is compared for male and female mice, but max rectal temperature is only shown for males. Was there a difference between males and females?

We added maximum rectal temperature of female mice in Fig. 1. By including data for additional experiments, we re-evaluated sex differences. There was no significant sex difference in maximum rectal temperature (**Fig. 1g**) and time to death (**Fig. 1h, 1i**). However, more male mice (90%) succumbed to isoflurane than females (65%) (**Fig. 1f**). Therefore, we used male mice for studies of the preventive effect of Cpd1 on isoflurane-induced MH crisis. These observations were described in Results (p. 5, lines 104-107; p. 7, lines 166-167).

2. Compound 1 causes a significant decrease in soleus force production from het mice. Were there any effects of Compound 1 on WT muscle force production?

We examined the effect of Cpd1 on WT muscle. It reduced twitch and tetanic contractions (**Fig. 3e, 3f**). These observations were described in Results (p. 7, lines 159-160).

3. What age mice are used? This should be stated in the methods and/or results section.

The ages of mice were described in methods section (p. 17, lines 405-406; p. 18, line 436; p. 19, lines 453, 467, and 475; p. 20, line 483).

4. The correct nomenclature should always be used when referring to the mouse strain instead of using the terms “het” and “hom”, especially since multiple mouse models are used. There’s also inconsistent use of “Wt” versus “WT”.

We now refer to the mouse strain as R2509C, R163C, and G2435R, instead of het or hom. We apologize mistake in expression of wild type. In the revised version WT was consistently used.

5. Line 121 –provide a citation for halothane stimulation of RyR1 and its use in MH testing.

We cited an article (Hopkins, Br J Anaesth, 2011) for the effect of halothane and its use in MH testing (p. 6, line 126).

6. Figure 2 – individual data points should be shown for panel D.

We added individual data points in the figure (**Fig. 3b**).

7. Figure 5 – Individual data points should be shown for panel D.

Since data points in this figure (**Fig. 8d**) are too many (n=39~78), data were given as box and whisker plots without individual data points.

8. Line 379 – change “x” to “X”

We changed "x" to "X" (p. 17, line 420).

9. There are many examples of incorrect wording and grammar throughout the text. Please proofread carefully and correct so that the text is easier to read. Some examples:

a. Line 160 – change “inducing” to “induction of”

b. Line 173 – change “of existing” to “in other”

c. Line 176 – change “was maintained a relatively” to “was maintained at a relatively”

- d. Line 182 – add “the” after tested
- e. Line 188 – change “at” to “in”
- f. Line 419 – add “at” after achieved
- g. Line 487 – change “points” to “point”.
- h. Line 667 – change “fully” to “full”

Thank you very much for your kind editing. We corrected all the points you suggested.

Reviewer #2 (Remarks to the Author):

Comments for Authors: Malignant hyperthermia (MH), a life-threatening disorder triggered by administration of volatile anesthetics (halothane, isoflurane, etc.), has been linked to mutations in the type-1 ryanodine receptor (RyR1), the Ca²⁺ release channel in skeletal muscle. Similar hyperthermic reactions (known as heat stroke; HS) can be triggered by exposure to high environmental temperature.

Dantrolene is the only approved drug for acute treatment of MH crises (though, not used in case of HS). Nevertheless, dantrolene has some disadvantages for clinical use: a) poor water solubility which makes its rapid preparation difficult in emergency situations; b) long plasma half-life, which causes side effects in patients (such as prolonged muscle weakness). No alternative drugs have been developed over 50 years to replace Dantrolene.

In this paper the authors created and characterized a new MH mouse model RYR1-p.R2509C; and identified in a oxolinic acid derivative (Compound 1, which was already reported as an inhibitor of RyR1 in 2019) a drug that effectively prevents and rescues MH and HS crisis triggered by isoflurane and heat in mice.

The results are important as they characterize a new drug which has the potential to replace dantrolene.

However, the paper may need some adjustments in the way data are currently presented.

Main Comments:

1. Fig. 2d and e: could the authors explain why they used Soleus muscle (a slow twitch muscle) for these IVCTs? Muscles in mice are mostly fast-twitch.

We examined the caffeine contracture of EDL muscles, but no apparent contracture was detected up to 20 mM caffeine (see the attached figure for reviewers). Soleus and diaphragm muscles, not EDL muscle, were used for contraction experiments of Y522S mutant mice (Chelu et al, 2004). It seems that mouse fast muscle is difficult to elicit caffeine contracture.

2. Figs. 2 and 3: could the authors explain why they used Halothane in Fig 2b and c, but Isoflurane in Fig. 3?

We carried out additional experiments of isoflurane on $[Ca^{2+}]_i$ in skeletal muscle cells (**Fig. 2e, 2f**). We found that isoflurane also increased $[Ca^{2+}]_i$ and Cpd1 completely prevent it. These observations were described in Results (p. 6, lines 133-135).

3. Comparison between Figs. 3 and 4: a. compound 1 was administered in one case when rectal temperature reached 39oC, but at 38oC in the second case: can the authors explain why?

According to reviewer's suggestion, we presented a figure in which Cpd1 was administered when rectal temperature reached 39°C at heat stress experiments (**Fig. 5e-5g**). The original Fig. 4 (administration at 38°C) was moved to **Supplementary Fig. 4** (p. 9, lines 203-209).

b. while Fig 3 is complete, it looks like Fig. 4 is INCOMPLETE, as in panels a-c the dose 3mg/kg is missing..... ??

In preliminary experiments, we found no effect of 3 mg/kg dose in heat stress experiments. So, we omitted the 3 mg/kg data. Since preventive effect is still weak even at 10 mg/kg, it is easily predicted that lower doses may have no or only minor effects.

c. It would be nice to have Figs.3 and 4 showing the same sets of data. Currently, panels b and c are different in the two Figures (while panel a and the 3 bottom panels are the same in both).

To show the same sets of data, we added maximum rectal temperature in heat stress experiments (**Fig. 5b**).

4. Fig. 5 shows in panel a) intracellular Ca²⁺ concentrations at rest, and in panel b) Mn²⁺ quench of fura-2 fluorescence. Nice to have those data included in the manuscript. However, they were collected in samples from MH mice different from the main knock-in mouse line used in the rest of the experiments. Why? Could this figure be moved into supplemental, and maybe rescue Suppl. Figures with data collected in the RYR1-p.R2509C knock-in mouse?

We carried out additional experiments of intracellular Ca²⁺ concentrations at rest and Mn²⁺ quench of fura-2 fluorescence of R2509C cells (**Fig. 2d and Supplementary Fig. 2**). These observations were described in Results (p. 6, lines 129-135, 136-140).

Minor Comments:

Title could be improved.

Suggestion 1: rescues volatile anesthesia-induced death and...

Suggestion 2: A novel RyR1 inhibitor prevents and rescues sudden death in a mouse model of malignant hyperthermia and heat stroke.

Thank you very much for your suggestion. We changed the title to "A novel RyR1 inhibitor prevents and rescues sudden death in a mouse model of malignant hyperthermia and heat stroke".

Introduction:

Page 3, line 55: Correct as follow: This MECHANISM is referred to as.....

We added "mechanism" at the site (p. 3, line 53).

Page 3, line 56: The RyR1 channel can also be directly activated by Ca²⁺.... I was under the impression that Ca²⁺ induced Ca²⁺ release was a mechanism used primarily by RyR2 and RyR3, while RyR1 is less sensitive to it.

We added a clause with one reference in the end of sentence: " although its importance in the function is much lower than RyR2 and RyR3 (Murayama and Kurebayashi, 2011) (p. 3, lines 55-56).

Results:

Page 5, line 104-105: Correct as follow: ...was similar among individuals MICE, and....

We corrected it (p. 8, line 194).

Page 5, line 114: Correct as follow: We initially tested the effect of Compound 1 IN in vitro experiments USING SINGLE FIBERS AND ISOLATED MUSCLES. Indeed, the authors in Figure 2 describe results in isolated FDB fibers and in whole Soleus muscles.

We inserted "in" before "in vitro experiments". We separated data for single cells and isolated muscles into two figures (**Figs. 2 and 3**) and the phrase was changed to "using enzymatically-isolated single flexor digitorum brevis (FDB) muscle cells from the R2509C mice" (p. 5, lines 120-121).

Page 5, line 124: Correct as follow: ... in het fibers, but not IN Wt....

Thank you for your editing. We corrected it (p. 6, line 134).

Page 6, line 129: ... of MH susceptibility in humans (please add REF). Please provide a reference after this statement.

We cited a reference (Larach et al., 1989) for IVCT test (p. 6, line 144).

Materials and Methods:

Page 16, line 388 & page 18, line 443: Remove Briefly, as the sections that follow are not brief.

We deleted "Briefly".

Figures:

Figure 1e, Y axis: should it be "maximum rectal temperature" as in Fig 3b?

We corrected the Y axis title of **Fig. 1d** to "maximum rectal temperature".

Figs. 3 and 4, panels a and d: a color/symbol legend (as - for example - in Figure 2d and e) would probably be better.

Compare Figs. 3e and 4e: at endpoint is missing

We showed rectal temperatures 10 min after administration and at endpoint. So, "at endpoint" was removed from the Y axis label (Fig. 4e, 5f).

Reviewers' Comments:

Reviewer #1:

Remarks to the Author:

The authors have done an excellent job addressing the issues of the original submission and the manuscript now provides a far more comprehensive assessment of the Cpd1. I have no further issues with the manuscript and recommend publication.

Reviewer #2:

Remarks to the Author:

The focus of the present paper is malignant hyperthermia (MH), a life-threatening disorder triggered by administration of volatile anesthetics (halothane, isoflurane, etc.). Dantrolene is the only approved drug for acute treatment of MH crises (though, not used in case of HS). Nevertheless, dantrolene has some disadvantages for clinical use: a) poor water solubility which makes its rapid preparation difficult in emergency situations; b) long plasma half-life, which causes side effects in patients (such as prolonged muscle weakness). No alternative drugs have been developed over 50 years to replace Dantrolene.

The results contained in this manuscript characterize a new drug which has the potential to replace dantrolene.

The paper is clearly written, and the authors have complied with all requests that this reviewer raised in the previous cycle of reviewing. Hence, I am completely satisfied by their revision of the manuscript.

At this stage, I have no additional comments for the authors.